# FLEXIBLELLM: MAKING LOW-BIT QUANTIZATION FOR LARGE LANGUAGE MODELS MORE FLEXIBLE AND EFFICIENT

## ABSTRACT

Low-bit quantization is crucial for deploying Large Language Models (LLMs) on resource-constrained hardware. However, existing Post-Training Quantization (PTQ) methods are limited by a monolithic view of outliers, failing to address their dual spatial distribution (both discrete and clustered) and overlooking "attribute outliers"—weights that are sensitive to quantization but not numerically large. Furthermore, these methods generally ignore the critical issue of quantization errors accumulating and amplifying across layers. To overcome these challenges, we introduce FlexibleLLM, a novel finetuning-free, weight-only PTQ framework founded on a new theoretical analysis of outliers. FlexibleLLM holistically addresses the outlier problem through three synergistic components: (1) To handle clustered outliers, the Self-Adaptive Block-Level Greedy Bit Search (SBGBS) module enables highly flexible, fractional-level bit-width allocation (e.g., 2.1 bits), optimizing the trade-off between hardware utilization and model accuracy. (2) For discrete outliers, the Discrete Outlier Suppression and Aware (DOSA) module employs a dual strategy: it innovatively uses Hadamard transforms for computationally efficient suppression of numerical outliers and a Hessian-aware mechanism to precisely handle overlooked "attribute outliers". (3) To combat error propagation, the Layer-Level Feedback and Denoising (LFD) module introduces a dynamic correction mechanism that mitigates the accumulation of "activation noise" from a global, cross-layer perspective. Extensive experiments demonstrate that FlexibleLLM achieves state-of-the-art performance, significantly outperforming not only existing finetuning-free methods but also many finetuning-based approaches, all while requiring substantially fewer computational resources. Code is available at https://anonymous.4open.science/r/FlexibleLLM.

## 1 INTRODUCTION

With the rapid advancement of large language models (LLMs) (Touvron et al., 2023b; Zhang et al., 2022), the demand among general users to independently deploy open-source LLMs on resource-constrained consumer-grade devices has become increasingly intense. To address this requirement, various efficiency-focused model compression techniques have been employed, including but not limited to pruning (Zhang et al., 2023), distillation (Gu et al., 2023), Quantization-Aware Training (QAT) (Du et al., 2024; Xu et al., 2024), and Post-Training Quantization (PTQ) (Guo et al., 2023; Li et al., 2025). Among these, PTQ has gained widespread attention due to its low cost.

Due to the limited memory capacity of consumer-grade devices, the community has an urgent demand for more aggressive low-bit quantization methods. However, under such configurations, the performance of quantized models degrades significantly (Frantar et al., 2022; Lin et al., 2024b). The widespread presence of outliers in weight matrices is a major factor constraining quantization performance (Zhang et al., 2024a). To address outliers, recent research(Lee et al., 2024; Huang et al., 2024a) has primarily drawn inspiration from the pruning methods (Frantar & Alistarh, 2023), identifying and mitigating outliers based on sensitivity scores of weights.

Despite significant progress, current PTQ methods face three primary challenges in handling outliers from local and global perspectives: **1) Local Outlier Misidentification.** Existing methods

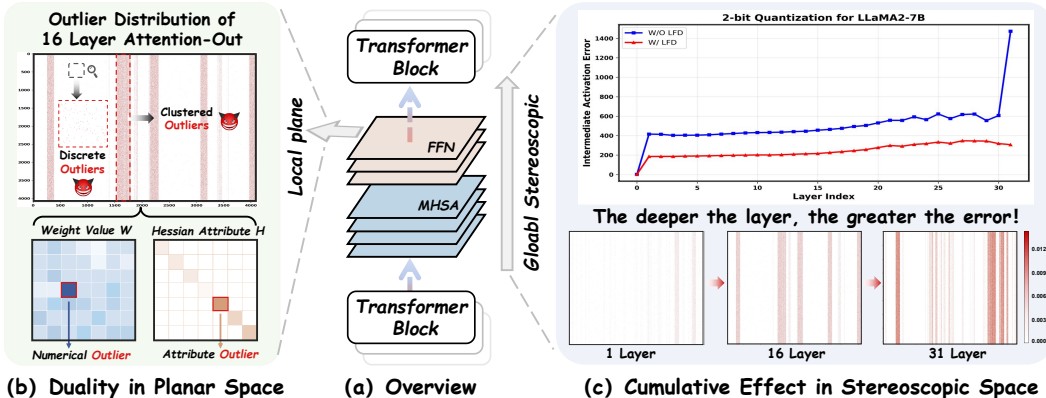

Figure 1: (a) The overview of multi-scale analytical perspective in FlexibleLLM. (b) illustrates the duality of outlier distribution in the planar space, and further distinguishes between numerical outliers and attribute outliers based on Hessian attribute values and element magnitudes. (c) reflects the cumulative effect of outliers and their impacts in the stereoscopic space. MHSA and FFN are the abbreviations of Multi-Head Self-Attention layer and Feed-Forward Network layer.

(Shao et al., 2023) narrowly define outliers as values with large magnitudes, overlooking a critical category of "attribute outliers": weights that have normal values but are highly sensitive to quantization and thus have a negative impact on model performance. As our analysis in Appendix C and Fig. 1 reveals, metrics like the Hessian diagonal can uncover a certain number of these attribute outliers. By focusing only on magnitude, conventional methods fail to detect these crucial elements, leading to their erroneous quantization and causing significant, irrecoverable accuracy degradation. **2) Local Inefficient Bit-width Allocation for Identified Outliers.** Even when outliers (typically high-magnitude ones) are correctly identified, current structured mixed-precision methods(Lee et al., 2024; Huang et al., 2024b) employ rigid, coarse-grained strategies. They assign a uniform high bit-width to entire regions containing outliers. This "one-size-fits-all" approach prevents fine-grained, importance-aware bit allocation, resulting in a suboptimal trade-off between accuracy and resource efficiency. **3) Myopic Optimization that Ignores Global Error Propagation.** Nearly all current methods(Sun et al., 2024; Guo et al., 2024) optimize quantization at the single-layer level, a "myopic" approach that ignores global error dynamics. They fail to account for that quantization errors from outliers may accumulate and amplify in the form of "activation noise" as they propagate across layers. This disregard for cross-layer dependencies means that locally optimal solutions do not guarantee a globally optimal model, often leading to significant overall performance degradation.

To address these challenges, we revisit the outlier problem from a novel multi-scale analytical perspective based on static sensitivity scores, revealing two key findings (detailed proofs in Appendix B): **1) Duality in Planar Space:** From a planar perspective (tensor level), the spatial distribution of outliers exhibits a duality characterized by the coexistence of discreteness and clustering, as shown in Fig. 1(b). This finding indicates that any strategy targeting only one of these distribution patterns may be suboptimal and cannot efficiently address both patterns simultaneously. **2) Cumulative Effect in Stereoscopic Space:** From a stereoscopic perspective (layer level), the impact of outliers is not independent across layers. Instead, as shown in Fig. 1(c), the sensitivity scores and quantization errors of each layer increase with the number of layers, and the quantization error even exhibits an exponential explosion in the last layer. This finding demonstrates the necessity of adopting a wider layer-level perspective to suppress the propagation of errors throughout the entire model.

Based on these findings, we propose a finetuning-free weight-only PTQ framework, FlexibleLLM. It jointly addresses the aforementioned three challenges through three core modules, namely the Self-Adaptive Block-Level Bit Search (SBGBS) module, Discrete Outlier Suppression and Awareness (DOSA) module, and Layer-Level Feedback Denoising (LFD) module. Specifically, **to address Challenge 1**, we design the DOSA module. Inspired by the duality in tensor-level space (Finding 1), this module processes discrete outliers from both numerical and intrinsic attribute perspectives without relying on unstructured separation methods, successfully overcoming the limitation of conventional magnitude-based detection approaches. **To address Challenge 2**, we develop the SBGBS module. This module introduces a greedy algorithm based on static sensitivity scores and dynamic

quantization errors to adaptively allocate optimal bit-widths to clustered elements, moving beyond the limitation of a small set of fixed bit options. Additionally, the SBGBS module enables flexible fractional-bit quantization (e.g., 2.1 bits), allowing full utilization of any tiny remaining storage space on hardware and maximizing resource efficiency. **To address Challenge 3**, we create the LFD module. Inspired by the cumulative effect in layer-level space (Finding 2), this module introduces a dynamic correction mechanism. By real-time comparing the difference in activation outputs between the full-precision model and the quantized model, it compensates for and denoises the "activation noise" accumulated layer by layer, effectively addressing the problem of layer-level myopia in existing methods. Through the coordinated operation of these three modules, FlexibleLLM provides a comprehensive solution to outlier-related challenges in PTQ, achieving significant improvements in both quantization accuracy and efficiency. Our contributions can be summarized as follows:

- We systematically diagnose the limitations of existing PTQ methods by pinpointing three critical challenges from both local and global perspectives: the misidentification of attribute outliers, inefficient bit-width allocation for identified outliers, and the neglect of cross-layer error propagation. This analysis provides a clear roadmap for addressing critical bottlenecks in low-bit quantization.

- We introduce a novel analysis of outliers, revealing two critical findings: the duality of outlier distributions (discrete and clustered) at the tensor level and the cumulative effect of quantization errors at the layer level. These insights establish the foundational principles for designing more effective quantization strategies.

- We propose a finetuning-free, weight-only PTQ framework that holistically addresses the identified challenges through three specialized modules: the DOSA module for accurate outlier identification, the SBGBS module for adaptive bit-width allocation, and the LFD module for suppressing cross-layer error propagation. The framework demonstrates significant improvements in quantization accuracy and efficiency.

## 2 RELATED WORK

### 2.1 POST TRAINING QUANTIZATION OF LLMS

Due to the presence of outliers, post-training quantization (PTQ) of large language models (LLMs) is a challenging task (Ma et al., 2024b). Existing PTQ methods for LLMs can be categorized based on whether activation are quantized into weight-only quantization (Lin et al., 2024b; Park et al., 2022) and weight-activation quantization (Lin et al., 2024a; Ma et al., 2024a; Liu et al., 2023). For weight-only quantization, some methods (Guo et al., 2024; Zhang et al., 2024a; Chee et al., 2023) attempt to mitigate numerical outlier through iterative optimization and linear transformations. Other recent studies (Zhang et al., 2024b; Yu et al., 2025) have employed low-rank decomposition to compensate for quantization errors in the weight matrix. Unlike weight-only quantization, weight-activation quantization need address outliers in both the weight matrix and activation matrix. Current weight-activation quantization methods primarily rely on linear transformation strategies to simultaneously mitigate outliers in both the weight and activation matrices. These strategies include finetuning-free online transformation methods (Ashkboos et al., 2024; Yi et al., 2024) and finetuning-based offline transformation methods (Liu et al., 2024; Hu et al., 2025). Given the massive parameter scale of LLMs, memory access efficiency becomes a primary bottleneck for acceleration. And weight-only quantization can effectively overcome this issue by compressing the weight matrix to low-bit representations. Therefore, our work primarily focuses on low-bit weight-only quantization.

### 2.2 FRACTIONAL-BIT QUANTIZATION

Fractional-level quantization is of great significance for the full utilization of hardware resources. Currently, some mixed-precision quantization methods inderictly achieve fractional-bit quantization by isolating outliers in the weight matrix and storing them in high precision. For example, SqueezeLLM (Kim et al., 2023) and SpQR (Dettmers et al., 2023) identify elements sensitive to quantization error and store them in full-precision format, achieving a quantization bit-width of 4.27 bits. In binarization scenarios, PB-LLM (Huang et al., 2024a) and BiLLM (Shang et al., 2023) store high-error elements in INT8, achieving 1.7-bit and 1.08-bit quantization respectively. However, these element-wise unstructured methods neglect the clustered distribution of outliers in the weight

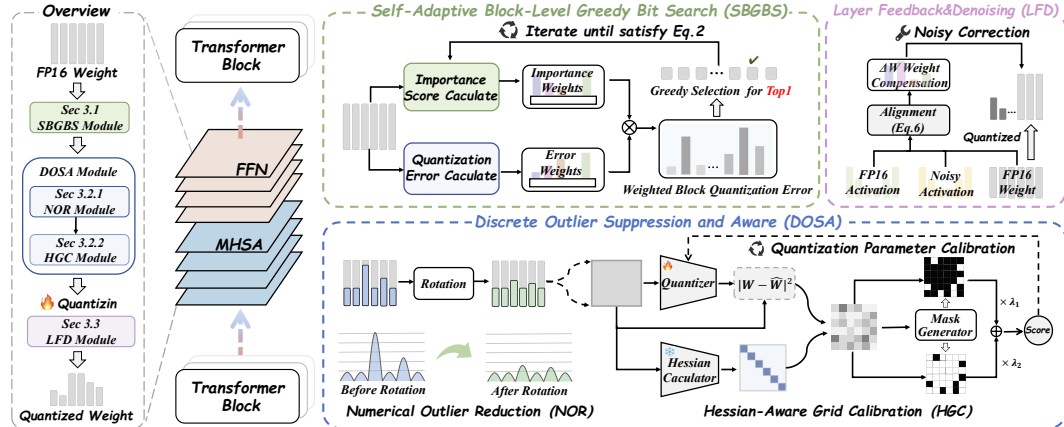

Figure 2: The overview of our FlexibleLLM. Given the FP16 weights as input, the Self-Adaptive Block-Level Greedy Bit Search (SBGBS) first identifies clustered outliers and assigns higher bit widths to them. Then, the Discrete Outlier Suppression and Awareness (DOSA) handles discretely distributed outliers from the perspectives of numerical magnitude and intrinsic attributes. Finally, the Layer-Level Feedback and Denoising (LFD) mitigates the "activation noise" generated during quantization from a wider, layer-level perspective, ultimately producing the quantized weights.

matrix and are unfriendly to hardware. Considering these factors, our approach utilizes a structured block-level greedy algorithm to adaptively adjust each block's bit-width based on its numerical fluctuations and intrinsic attributes within a given target bit-width.

## 3 METHODS

In this section, we provide a overview of FlexibleLLM. The overall framework is shown in Fig. 2. Specifically, in Sec.3.1, we fully focus on the clustered outliers and achieve flexible fractional-bit quantization by introducing the Self-Adaptive Block-Level Greedy Bit Search module. Subsequently, in Sec.3.2, we propose the Discrete Outlier Suppression and Aware module to handle discretely distributed outliers from two perspectives: numerical magnitude and intrinsic attributes. Finally, in Sec.3.3, based on the "activation noise" phenomenon caused by outliers, we introduce the Layer-Level Feedback and Denoising module to denoise during the quantization process.

### 3.1 SELF-ADAPTIVE BLOCK-LEVEL GREEDY BIT SEARCH

To break through the performance bottleneck of quantized models, we analyzed the spatial distribution characteristics of outliers in weight matrices. As illustrated in Fig. 1, we observed outliers exhibit two types of spatial distributions: continuous-column clustered and discrete. This finding prompted us to design different strategies to handle outliers with these two distribution patterns. For outliers with clustered distribution, we propose a Self-Adaptive Block-Level Greedy Bit Search (SBGBS) algorithm to allocate more bit-widths to clustered outliers in a flexible manner.

Specifically, we first divide the weight matrix into blocks with 128 columns as a unit, and then calculate the importance score for each block. In SparseGPT (Frantar & Alistarh, 2023), the importance score of a single element in the weight matrix is defined as $\frac{w_{i,j}^2}{[H^{-1}]_{j,j}^2}$. Inspired by this, the calculation formula for the block importance score of the SBGBS module is as follows:

$$B = \sum_{i=1}^{n} \sum_{j=1}^{m} \frac{w_{i,j}^2}{[H^{-1}]_{j,j}^2} \tag{1}$$

where $i$ and $j$ denote the row and column indices of the weight matrix. $n$ and $m$ represent the number of rows and columns in each block-level unit, and $H^{-1}$ denotes the inverse Hessian matrix.

Block importance $B$ reflects the sensitivity of weight blocks to quantization error. The SBGBS module integrates static block importance scores and dynamic quantization loss with weighting to

create a greedy optimization target, ensuring that each optimization step prioritizes weight blocks most sensitive to quantization error and with the most prominent numerical fluctuations. The greedy optimization target and constraints in the SBGBS module are defined as follows:

$$\text{Obj}: \underset{b_1,\ldots,b_n}{\arg\min} \sum_{i=1}^{n} (B_i)^{\alpha} \cdot ||W_i - quant_{b_i}(W_i)||_2^2$$
$$\text{Constrain}: \frac{1}{n}\sum_{i=1}^{n} b_i \leq target\_bit \tag{2}$$

where $W_i$, $B_i$ and $b_i$ denote the numerical matrix, importance score, and bit-width of the $i$-th weight block. $\alpha$ controls the impact of block importance scores on the greedy process, which we set to 2.

The SBGBS module will carry out the entire greedy iterative process based on three main steps until the average bit-width reaches the preset target bit-width: **1) Initialization:** Firstly, given a preset target bit-width, SBGBS module will assign the same initial bit-width to each weight block, which is typically set to 1 or 2 bit. **2) Greedy optimization:** SBGBS module assume that the quantization bit-width of all weight blocks is increased by 1, and then calculate the reduction in weighted quantization loss of each block based on Eq (2). **3) Greedy selection:** SBGBS module select the block with the largest reduction in weighted quantization loss to increase its actual quantization bit-width.

It is worth noting that in the SBGBS module, the preset target bit-width can not only be an integer, but also be extended to the fractional level. This unique advantage enables users to flexibly adjust the quantization bit-width based on the actual storage space of hardware devices, so as to maximize the storage space utilization and the performance of the quantized model.

### 3.2 Discrete Outlier Suppression and Aware

After addressing the clustered outliers, we turned our focus to discrete outliers. Specifically, we introduce the Discrete Outlier Suppression and Aware module to handle discrete outliers in weight matrices from numerical magnitude and intrinsic attribute. It consists of two components: the Numerical Outlier Reduction (NOR) module and the Hessian-Aware Grid Calibration (HGC) module.

### 3.2.1 Numerical Outlier Reduction.

For discrete numerical outliers, We note that prior work (Ma et al., 2024a; Sun et al., 2024) has attempted to mitigate them by obtaining optimal transformation matrices based on fine-tuning. While such approaches are computationally demanding, they have inspired us to address numerical outliers from the perspective of linear transformations. Inspired by early mathematical research (Sylvester, 1867), we observe that the Hadamard transform can effectively amortize the outliers into other entries. Additionally, Hadamard transformation matrices can be rapidly constructed through recursive methods, requiring minimal computational resources. Therefore, we adopt Hadamard matrix tailored to the hidden dimensions of weight to perform transformation operations:

$$Y = WX = (WH)(H^{-1}X) \tag{3}$$

where $H$ is Hadamard matrix and satisfies the orthogonality property $H = H^{-1}$. This operation leverages the principle of computational invariance to eliminate discrete numerical outliers and reshape a balanced data distribution without altering the model's output.

### 3.2.2 Hessian-Aware Grid Calibration.

In this module, we leverage the Hessian attributes of weight elements to derive the optimal quantization parameters for handling "attribute outliers". Specifically, we first perform quantization based on the existing quantization parameters. Then, we compute the weighted error matrix $\mathcal{L}$ for the weight block by integrating the Hessian information with the mean squared error loss between the original weight $W$ and the quantized weight $\hat{W}$:

$$\mathcal{L} = \left( diag\left(H^{-1}\right)^{-P} \right)^{\top} \left\| quant\left(\hat{W}, s, z\right) - W \right\|_2^2 \tag{4}$$

where $P$ a hyperparameter that controls the influence degree of the Hessian attributes on parameter calibration, which we set to 2. $s$ and $z$ are scaling factor and zero point respectively.

Subsequently, we apply the 3-$\sigma$ rule to generate relevant masks $M_o$ and $M_u$ based on the weighted error matrix $\mathcal{L}$. These masks enable us to separate the processing of attribute outliers from that of normal elements. To obtain the optimal scaling factor $s$ and zero point $z$, gradient descent could be employed in theory, but this approach tends to require high costs in terms of time and memory. We therefore search for $s$ and $z$ over a constrained grid and formulate the optimization as:

$$\arg\min_{s,z} \lambda_1 \cdot \mathcal{L}_u(s,z) + \lambda_2 \cdot \mathcal{L}_o(s,z), \text{ where}$$
$$s = \left(l + t\frac{R}{T}\right)\frac{\max(W) - \min(W)}{2^b - 1}, z = -\left[\frac{\left(l + t\frac{R}{T}\right)min(W)}{s}\right] \tag{5}$$

Here, $\mathcal{L}_u(s,z)$ and $\mathcal{L}_o(s,z)$ denote the weighted error corresponding to normal elements and outlier elements respectively. They are obtained by element-wise multiplying $\mathcal{L}$ with the masks $M_o$ and $M_u$. $\lambda_1$ and $\lambda_2$ are the loss weighting coefficients, which we set to 1 and 1.2. $R$ represents the range of grid optimization, $l$ is the left endpoint of grid optimization range, $T$ is the search step size, and $b$ is the quantization bit-width. To balance efficiency and accuracy, we empirically set $R$, $l$ and $T$ at 0.8, 0.2 and 100.

### 3.3 LAYER-LEVEL FEEDBACK AND DENOISING

As shown in Fig. 1(c), the quantization error propagates layer by layer, exhibiting exponential growth in the final layer—a phenomenon which we attribute to the accumulation of "activation noise". While end-to-end tuning could theoretically mitigate this issue, its prohibitive computational cost hinders practical deployment. Moreover, most existing quantization methods rely on the quantized model's own activations for optimization, a strategy that inadvertently perpetuates error propagation across layers and further exacerbates the accumulation of activation noise. Inspired by the negative feedback mechanism in automatic control principles (Franklin et al., 2002), We utilize the full precision intermediate activation values as the learning target to mitigate the accumulated "activation noise" phenomenon. The overall alignment objective is formulated as follows:

$$\underbrace{\arg\min_{\widehat{W_\ell}} ||W_\ell X_\ell - \widehat{W_\ell} X_\ell||_2^2}_{Original} \quad \Rightarrow \quad \underbrace{\arg\min_{\widehat{W_\ell}} ||W_\ell X_\ell - \widehat{W_\ell}\widehat{X_\ell}||_2^2}_{FlexibleLLM(Ours)} \tag{6}$$

where $W_\ell$ and $X_\ell$ represent the full-precision weight matrix and the input activation of the $\ell$-th layer. $\widehat{W_\ell}$ and $\widehat{X_\ell}$ represent the quantized weight matrix and the input activation of the $\ell$-th layer.

Meanwhile, inspired by the OBQ method (Frantar & Alistarh, 2022), we try to correct the unquantized elements in real-time when quantizing weight elements column by column, so as to reduce the activation noisy. Specifically, during the quantization process, after quantizing the $i$-th column elements $w_i$, we need to correct the remaining unquantized weight columns to minimize the alignment optimization objective. We assume this correction amount is $\delta_i$. To solve for $\delta_i$, we first transform the alignment optimization objective into an optimization expression related to $\delta_i$:

$$\arg\min_{\delta_i} ||W_\ell(X_\ell - \widehat{X_\ell}) - \delta_i \widehat{X_\ell}||_2^2 \tag{7}$$

Then, by constructing the constraint condition $\delta_i e_i^\top - (quant(w_i) - w_i) = 0$ ($e_i$ is a one-hot vector with 1 at the i-th position), we transform the problem of solving $\delta_i$ into an optimization problem solvable by the Lagrangian function:

$$E(w_i) = e_i^\top \delta_i - (quant(w_i) - w_i)$$
$$L(\delta_i, \lambda) = ||W_\ell(X_\ell - \widehat{X_\ell}) - \delta_i \widehat{X_\ell}||_2^2 + \lambda E(w_i) \tag{8}$$

Next, we take the derivatives of $L(\delta_i, \lambda)$ with respect to $\delta_i$ and $\lambda$ and set the derivative results to 0.

$$\begin{cases} 2(W_\ell(X_\ell - \widehat{X_\ell}) - \delta_i\widehat{X_\ell})(-\widehat{X_\ell}) + \lambda e_i = 0 \\ e_i^\top \delta_i - (quant(w_i) - w_i) = 0 \end{cases} \tag{9}$$

Combined with the Hessian formula $H = 2\widehat{X_\ell}\widehat{X_\ell}^\top$, we can obtain:

$$\delta_i = \frac{quant(w_i) - w_i}{H_{i,i}^{-1}} H_{:,i}^{-1} + W_\ell(X_\ell - \widehat{X_\ell})\widehat{X_\ell}^\top H_{-i}^{-1} \tag{10}$$

where $\delta_i$ is the correction amount for full precision unquantized elements in weight matrix to minimum the alignment optimization objective. $H_{-i}^{-1}$, represents the inverse of the Hessian matrix with the $i$-th row and $i$-th column removed.

Considering the memory throughput issue, we adopt a batch strategy with $B = 128$ columns to correct the unquantized weight elements. Specifically, we quantize weight elements column by column. Before the quantization of a batch ends, we limit the correction within the range corresponding to $row \times B$ of this batch. After the quantization of this batch ends, we use the following formula to compensate for the unquantized elements outside this batch:

$$E_{corr} = W_Q(\left(X - \hat{X}\right)\hat{X}^\top H_F^{-1})_{:,Q}$$
$$\delta_F = \frac{quant(W_Q) - W_Q}{\left[H_F^{-1}\right]_{QQ}}\left(H_F^{-1}\right)_{:,Q} + E_{corr} \tag{11}$$

where $Q$ denotes the set of indices of the quantized weight elements, and $F$ represents the set of the remaining unquantized weights. $H_F^{-1}$ represents the inverse of the Hessian matrix with the corresponding rows and columns removed.

## 4 EXPERIMENTS

### 4.1 EXPERIMENTAL SETUP

**Models and Baselines.** We apply FlexibleLLM to pretrained LLMs including LLaMA (Touvron et al., 2023a), LLaMA-2 (Touvron et al., 2023b) and LLaMA-3 (Dubey et al., 2024) family. These models are representative open-source LLMs, enabling research across diverse model scales and architectures. We compared our method against both finetuning-free and finetuning-based weight-only PTQ methods. For finetuning-free PTQ methods, we compare FlexibleLLM with Round-To-Nearest(RTN), GPTQ (Frantar et al., 2022), AWQ (Lin et al., 2024b), LeanQuant (Zhang & Shrivastava, 2024), SliM-LLM (Huang et al., 2024b), QuIP (Chee et al., 2023), PB-LLM (Shang et al., 2023). For finetuning-based PTQ methods, we compare FlexibleLLM with OmniQuant (Shao et al., 2023), AffineQuant (Ma et al., 2024a) and decoupleQ (Guo et al., 2024).

**Datasets and Evaluation.** Firstly, We evaluate the language modeling capabilities of quantized models on language generation tasks based on the WikiText2 datasets (Merity et al., 2016). Then we conduct experiments on zero-shot evaluation tasks including Winogrande (Sakaguchi et al., 2021), PIQA (Bisk et al., 2020), ARC-c, ARC-e (Boratko et al., 2018) and HellaSwag (Zellers et al., 2019).

**Experimental Details.** We implement FlexibleLLM based on Huggingface (Wolf et al., 2019) and PyTorch (Paszke et al., 2019). All experiments were conducted on NVIDIA A800 80GB GPUs.

### 4.2 MAIN RESULTS

We firstly test the performance of FlexibleLLM on the LLaMA family. As shown in Table 1, our method consistently outperforms previous state-of-the-art (SOTA) methods across all configurations and models. Specifically, in language generation tasks, compared with the best baseline method, our method achieves an average performance improvement of 3.59 points and 0.12 points under 2-bit and 3-bit quantization configurations. On the most challenging LLaMA3-8B model, FlexibleLLM achieves an improvement of 28.1 points in the perplexity metric compared with the best baseline method under the 2-bit quantization configuration. In zero-shot tasks, our method achieves an average performance improvement of 3.37 points and 2.59 points under 2-bit and 3-bit quantization configurations compared with the best baseline method. These significant performance improvements demonstrate the effectiveness of our method. It is worth noting that FlexibleLLM is a finetuning-free PTQ method, which requires fewer computational resources to achieve better quantization performance compared with the finetuning-based PTQ methods in Table 1.

Table 1: Main results of evaluation experiment. We report the perplexity and zero-shot accuracy. The best score is **bolded**.

| Bits | Method | Tuning-free | WikiText2 perplexity ↓ | | | | | | | | | Zeroshot avg acc. ↑ | | | | | | |
|---|---|---|---|---|---|---|---|---|---|---|---|---|---|---|---|---|---|---|
| | | | 7B | 13B | 30B | 65B | 2-7B | 2-13B | 2-70B | 3-8B | 3-70B | 7B | 13B | 30B | 2-7B | 2-13B | 3-8B | 3-70B |
| 16 | FP16 | - | 5.68 | 5.09 | 4.10 | 3.53 | 5.47 | 4.88 | 3.31 | 5.75 | 2.9 | 68.6 | 70.96 | 74.52 | 69.18 | 71.75 | 72.87 | 79.94 |
| 3.0 | RTN | ✓ | 7.01 | 5.88 | 4.87 | 4.24 | 6.66 | 5.51 | 3.97 | 29.91 | 11.84 | 47.99 | 45.99 | 51.77 | 62.06 | 65.77 | 58.72 | 65.29 |
| 3.0 | GPTQ | ✓ | 6.55 | 5.62 | 4.80 | 4.17 | 6.29 | 5.42 | 3.85 | 8.19 | 5.22 | 39.65 | 54.63 | 57.76 | 62.48 | 66.18 | 60.58 | 71.28 |
| 3.0 | AWQ | ✓ | 6.46 | 5.51 | 4.63 | 3.99 | 6.24 | 5.32 | - | 8.22 | 4.81 | 32.26 | 35.44 | 35.07 | 62.82 | 66.14 | 64.82 | 73.65 |
| 3.0 | SliM-LLM | ✓ | 6.40 | 5.48 | 4.61 | 3.99 | 6.24 | 5.26 | 3.67 | 7.16 | 4.08 | - | - | - | - | - | - | - |
| 3.0 | LeanQuant | ✓ | 6.62 | 5.76 | - | - | 6.61 | 5.66 | 3.91 | 7.88 | - | 63.85 | 64.88 | - | 61.06 | 65.18 | 54.54 | - |
| 3.0 | OmniQuant | ✗ | 6.15 | 5.44 | 4.56 | 3.94 | 6.03 | 5.28 | 3.78 | - | - | 63.89 | 67.91 | 72.04 | 62.42 | 66.18 | 64.09 | 71.90 |
| 3.0 | AffineQuant | ✗ | 6.14 | 5.45 | 4.59 | - | 6.08 | 5.28 | - | - | - | 63.82 | 67.58 | 72.45 | 64.02 | 67.68 | 60.26 | - |
| 3.0 | decoupleQ | ✗ | 6.38 | 5.60 | 4.67 | 6.05 | 6.22 | 5.72 | 3.84 | - | - | - | - | - | - | - | - | - |
| 3.0 | FlexibleLLM | ✓ | **6.02** | **5.40** | **4.46** | **3.81** | **5.83** | **5.17** | **3.61** | **7.08** | **4.01** | **66.66** | **69.68** | **73.25** | **66.73** | **69.20** | **69.62** | **77.39** |
| 2.0 | RTN | ✓ | 1.9e3 | 781.2 | 68.04 | 15.08 | 4.2e3 | 122.08 | 27.27 | 1.9e3 | 4.6e5 | 35.97 | 35.12 | 35.98 | 35.60 | 35.56 | 35.87 | - |
| 2.0 | GPTQ | ✓ | 152.31 | 20.44 | 13.01 | 9.51 | 60.45 | 28.14 | 8.78 | 210 | 11.9 | 36.88 | 55.24 | 56.90 | 46.31 | 40.49 | 40.12 | 45.42 |
| 2.0 | AWQ | ✓ | 2.6e5 | 2.8e5 | 2.4e5 | 7.4e4 | 2.2e5 | 1.2e5 | - | 1.7e6 | 1.7e6 | 32.47 | 36.35 | 35.28 | - | - | - | - |
| 2.0 | QuIP | ✓ | 29.74 | 12.48 | 11.57 | 7.83 | 39.73 | 13.48 | 6.64 | 84.97 | 13.03 | - | - | - | - | - | - | - |
| 2.0 | PB-LLM | ✓ | 24.61 | 17.73 | 12.65 | 7.85 | 25.37 | 49.81 | NAN | 44.12 | 11.68 | - | - | - | - | - | - | - |
| 2.0 | SliM-LLM | ✓ | 14.58 | 8.87 | 7.33 | 5.90 | 16.01 | 9.41 | 6.28 | 39.66 | 9.46 | 41.14 | 56.45 | 59.25 | - | - | - | - |
| 2.0 | LeanQuant | ✓ | 18.53 | 14.42 | - | - | 25.59 | 24.43 | 7.92 | 41.78 | - | 49.86 | 53.38 | - | 42.43 | 48.00 | 38.98 | - |
| 2.0 | OmniQuant | ✗ | 9.71 | 7.93 | 7.12 | 5.95 | 11.06 | 8.26 | 6.55 | - | - | 47.32 | 56.68 | 59.87 | 46.98 | 53.56 | 52.66 | 60.06 |
| 2.0 | AffineQuant | ✗ | 13.51 | 7.22 | 6.49 | - | 10.87 | 7.64 | - | - | - | 48.55 | 52.56 | 62.00 | 37.20 | 49.50 | 36.03 | - |
| 2.0 | decoupleQ | ✗ | 9.49 | 7.86 | 6.37 | 5.59 | 9.74 | 13.03 | 5.23 | - | - | - | - | - | - | - | - | - |
| 2.0 | FlexibleLLM | ✓ | **8.82** | **6.82** | **5.84** | **5.00** | **8.70** | **6.94** | **4.87** | **11.56** | **9.32** | **51.02** | **60.36** | **65.99** | **52.7** | **59.08** | **54.92** | **61.32** |
| 2.1 | FlexibleLLM | ✓ | **7.76** | **6.47** | **5.61** | **4.85** | **7.72** | **6.51** | **4.70** | **10.68** | **7.96** | **55.39** | **62.71** | **66.85** | **56.72** | **60.91** | **56.09** | **67.55** |

Table 2: Ablation study of main components in FlexibleLLM. The evaluation was conducted by removing individual components and measuring the WikiText2 perplexity.

| SBGBS | NOR | HGC | LFD | WikiText2 perplexity ↓ | | | |
|---|---|---|---|---|---|---|---|
| | | | | 1-7B | 1-13B | 2-7B | 2-13B |
| ✗ | ✓ | ✓ | ✓ | 9.68 | 7.37 | 10.17 | 7.81 |
| ✓ | ✗ | ✓ | ✓ | 10.33 | 9.92 | 15.35 | 8.80 |
| ✓ | ✓ | ✗ | ✓ | 11.31 | 8.82 | 12.01 | 9.20 |
| ✓ | ✓ | ✓ | ✗ | 14.40 | 7.99 | 15.50 | 8.8 |
| ✓ | ✓ | ✓ | ✓ | **8.82** | **6.82** | **8.70** | **6.94** |

Table 3: Comparison of Time (hour) and Memory (GB) Costs During Quantization with Finetuning-Based PTQ Methods. Our approach consistently achieves the lowest computational.

| method | 2-7B | | 2-13B | | 2-70B | |
|---|---|---|---|---|---|---|
| | Time | Mem | Time | Mem | Time | Mem |
| OmniQuant | 7.27 | 14.35 | 10.68 | 19.91 | 24.88 | 39.69 |
| AffineQuant | 7.11 | 23.49 | 11.17 | 30.60 | 23.92 | 130.26 |
| Ours | **0.56** | **10.39** | **0.93** | **13.53** | **4.58** | **31.32** |
| Ours(2.1bit) | **0.57** | **10.39** | **0.97** | **13.54** | **5.31** | **31.35** |

Furthermore, we found that in low-bit quantization scenarios, for quantized models, a slight increase in the quantization bit-width can significantly improve their performance. For example, the 2.1-bit quantized LLaMA3-70B version of FlexibleLLM achieves an improvement of 14.6% and 10.2% in perplexity and zero-shot accuracy compared with its 2-bit quantized version. This indicates that in low-bit quantization deployment scenarios, if hardware devices have available storage capacity, appropriately increasing the bit-width of the quantized model using FlexibleLLM can substantially enhance the model performance.

## 4.3 ABLATION STUDY

**Effectiveness of Individual Components.** In this section, we conduct a detailed ablation study to evaluate the impact of each component on the performance of FlexibleLLM. As shown in Table 2, we observe that when removing the HBGB, NOR, and HGC modules, the perplexity of our method on different LLMs increases by 0.55-1.47 points, 1.51-6.65 points, and 2-3.11 points respectively. This indicates that effectively focusing on the spatial distribution patterns and intrinsic attributes of outliers can enhance quantization performance. Meanwhile, we note that when the LFD module is removed, the perplexity of our method rises by 1.17-6.8 points. This further demonstrates the significance of addressing the "activation noise" caused by outliers from layer-level perspective.

**Results of Instruction Following Models.** To evaluate its instruction-following capability, we quantized the Llama3-8B-Chat model and evaluated its performance using the Vicuna benchmark (Chiang et al., 2023) and GPT-4 evaluation protocol (Chiang et al., 2023). Previous study (Zheng et al., 2023) has identified an order bias in GPT-4 scoring, where models presented first tend to receive higher ratings. To mitigate this bias in the GPT-4-based evaluation protocol, we conduct 160 pairwise comparisons using two permutation orders and aggregate the results. As shown in Fig. 3, under 2-bit quantization scenarios, FlexibleLLM achieved win-tie rates of 91.9%, 78.8% and 81.9% against GPTQ, OmniQuant and AffineQuant, respectively. These results demonstrate that FlexibleLLM exhibits strong capability in quantizing instruction-finetuned LLMs.

Table 4: Perplexity performance on WikiText2 datasets of different structured bit allocation strategies on LLaMA model families.

| method | Bits | 1-7B | 1-13B | 1-30B | 1-65B |
|---|---|---|---|---|---|
| OWQ | 3.1 | 6.41 | 5.66 | 4.75 | 4.25 |
| SLIM-LLM | 3.0 | 6.40 | 5.48 | 4.61 | 3.99 |
| SBGBS | 3.0 | **6.29** | **5.44** | **4.55** | **3.93** |

Table 5: Perplexity on WikiText2 datasets of Qwen model families when quantized to 2 and 3 bit by different methods.

| method | Qwen1.5-7B | | Qwen2-7B | | Qwen2.5-7B | |
|---|---|---|---|---|---|---|
| | 2Bit | 3Bit | 2Bit | 3Bit | 2Bit | 3Bit |
| GPTQ | 37.34 | 9.28 | 23.61 | 7.85 | 25.8 | 7.98 |
| OmniQuant | 29.02 | 9.18 | 14.39 | 7.96 | 14.73 | 7.87 |
| Ours | **11.99** | **8.55** | **10.94** | **7.59** | **10.17** | **7.41** |

Figure 3: GPT-4 evaluation on the Vicuna.

Figure 4: Inference Speed Comparison.

**Results of Different Structured Bit Allocation Strategies.** We compared two sensitivity-based structured bit allocation algorithms with the SBGBS module on the LLaMA model, using the GPTQ quantization framework. As shown in Table 4, SBGBS consistently achieves optimal performance. This can be attributed to two key factors: 1) It comprehensively considers both static sensitivity scores and dynamic quantization errors during bit-width allocation. 2) It adaptively assigns the most suitable bit-width to each region, without being constrained by predefined bit-widths.

## 4.4 EFFICIENCY AND GENERALIZATION ANALYSIS

**Quantization Speed and Peak Memory.** One of the significant advantages of FlexibleLLM is that it is a finetuning-free method, which requires less computational resources compared to finetuning-based methods. To further demonstrate this advantage, we compare the quantization time and peak memory between FlexibleLLM and existing finetuning-based quantization methods in Table 3. Combined with results from Table 1, we can conclude that FlexibleLLM can achieve superior quantization performance at the cost of lower resource overhead.

**Results of Inference Efficiency.** Due to the potential inference latency that may be introduced by the mixed-precision quantization method in FlexibleLLM, we compared the inference speed differences among FlexibleLLM, GPTQ, and the full-precision model in Fig. 4. The results show that the inference speeds of the 2-bit Llama2-7B and Llama2-13B models quantized by FlexibleLLM are 1.31x and 3.27x those of the full-precision models. Moreover, the inference speeds of our method on these two models are only 4.4% and 2.4% slower than those of the lighter GPTQ method. In addition, we also tested the inference speed of FlexibleLLM in the 2.1-bit quantization scenario, and the results show that the additionally allocated fractional-level bit-widths have nearly negligible impact on the inference performance.

**Results of Other Model Families.** To evaluate the effectiveness of FlexibleLLM in quantizing other model families, we selected the Qwen series of LLMs as new quantization targets. Table 5 presents the performance of Qwen1.5-7B, Qwen2-7B and Qwen2.5-7B after being quantized by FlexibleLLM. These results indicate that the quantization capability of FlexibleLLM is not limited to LLMs with a specific architecture.

## 5 CONCLUSION

In this study, we propose FlexibleLLM, a finetuning-free weight-only PTQ method to make low-bit quantization more efficient and flexible by focusing on the spatial distribution and intrinsic attributes of outliers as well as the "activation noise" they introduce. The SBGBS module in FlexibleLLM enables flexible fractional-bit quantization based on predefined bit-width sets, and can adaptively allocate more bit-width to outliers with clustered distributions. Meanwhile, the DOSA module handles outliers with discrete distributions from both numerical magnitude and intrinsic attribute perspectives. Additionally, the LFD module performs feedback calculation and denoising compensation on noise errors caused by outliers from Layer-Level perspective. Extensive experiments demonstrate that FlexibleLLM outperforms existing PTQ methods across various LLMs and benchmarks, while balancing inference efficiency and quantization speed without requiring additional finetuning.

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

# A    THE USE OF LARGE LANGUAGE MODELS

We utilized Large Language Models to aid and polish the writing of this paper. Specifically, Large Language Models are used to refine language expression, boost sentence clarity and fluency, and ensure logical text coherence. We are grateful for the valuable contribution Large Language Models have made to the improvement of this paper's writing.

# B    SPATIAL DISTRIBUTION PATTERNS OF OUTLIERS

**Theorem 1.** For LLMs, given an activation value matrix $X \in R^{t \times m}$, a weight matrix $W \in R^{n \times m}$ and a Hessian matrix $H = XX^\top$ of a certain layer. If the sensitivity evaluation index $s_{i,j} = \frac{w_{i,j}^2}{[H^{-1}]_{j,j}^2}$ is used as the evaluation index for outliers, then the outliers exhibit clustering by continuous columns and discrete distribution in the matrix space.

**Proof 1.** Given an arbitrary column index set $S = \{q, q+1, ..., q+K+1\}$, for the columns inside the set S and those outside the set, there exist a lower bound $\tau$ and an upper bound $\beta$ for the Euclidean norm respectively, that is,

$$||X_{:,j}||_2 \geq \tau \ (\forall j \in S), ||X_{:,k}||_2 \leq \beta \ (\forall k \notin S) \tag{12}$$

We set the lower and upper bounds of the absolute values of the weight amplitudes within the index S as $w_{\min}$ and $w_{\max}$. That is the following holds:

$$0 < w_{\min} \leq |w_{i,j}| \leq w_{\max}(\forall j \in S, i) \tag{13}$$

Since $H = XX^\top$, it follows that:

$$s_{i,j} = \frac{w_{i,j}^2}{[H^{-1}]_{j,j}^2} = \frac{w_{i,j}^2}{\left[(XX^\top)^{-1}\right]_{j,j}^2} = w_{i,j}^2 ||X_{:,j}||_2^4 \tag{14}$$

For any $j \in S$ and arbitrary i:
$$s_{i,j} \geq w_{\min}^2 \tau^4 \tag{15}$$

For any $k \notin S$ and arbitrary i:
$$s_{i,k} \leq w_{\max}^2 \beta^4 \tag{16}$$

If
$$w_{\min}^2 \tau^4 > w_{\max}^2 \beta^4 \Leftrightarrow \frac{\tau}{\beta} > \left(\frac{w_{\max}}{w_{\min}}\right)^{\frac{1}{2}} \tag{17}$$

then, from (15) and (16), it follows that for any $k \notin S$, arbitrary $j \in S$, and arbitrary i,

$$s_{i,j} > s_{i,k} \tag{18}$$

Under the condition (18), performing an element-wise comparison across a whole row immediately yields a column-wise ordering: for all columns $j \in S$, their sensitivity element values are strictly greater than those of columns in the complementary set. Since S itself is by definition a continuous interval of indices, the highly sensitive elements must necessarily appear as a consecutively ordered collection of columns.

We then take the weight element $w_{m,n}$ from the m-th row and n-th column in the vicinity of $w_{i,j}$. Its sensitivity score is

$$s_{m,n} = \frac{w_{m,n}^2}{[H^{-1}]_{n,n}^2} = w_{m,n}^2 H_{n,n}^2 \tag{19}$$

i) If j $\neq$ n, then

$$\frac{s_{i,j}}{s_{m,n}} = \frac{w_{i,j}^2 H_{j,j}^2}{w_{m,n}^2 H_{n,n}^2} \tag{20}$$

In this case, both the weight magnitude and the corresponding diagonal elements of the Hessian jointly determine the difference in saliency scores between $w_{i,j}$ and $w_{m,n}$.

ii) If j = n, then

$$\frac{s_{i,j}}{s_{m,n}} = \frac{w_{i,j}^2 H_{j,j}^2}{w_{m,n}^2 H_{j,j}^2} = \frac{w_{i,j}^2}{w_{m,n}^2} \tag{21}$$

In this case, the weight magnitude alone determines the difference in saliency scores between $w_{i,j}$ and $w_{m,n}$.

Combining (i) and (ii), we can conclude that for the element $w_{i,j}$, if its corresponding Hessian diagonal entry and weight magnitude are both relatively large, then its sensitivity score will be significantly greater than those of the elements in the vicinity of $w_{i,j}$. As a result, in the overall distribution space, it forms an anomalous value that exhibits a discrete distribution pattern.

**Thoerem 2.** Suppose a large language model consists of L layers. As the quantized network goes deeper, the sensitivity scores of the elements in each layer grow progressively with the increase of the layer index.

**Proof 2.** Given the activation matrices of layer $l + 1$ and layer $l$ as $X_{l+1}$, $X_l \in R^{\mathrm{t} \times \mathrm{m}}$. The Hessian matrix is $H = XX^\top$. $\sigma$ denotes the activation function of the linear layer. $\varphi$ denotes the singular value of the corresponding matrix. The forward propagation process of the activation matrix of layer $l$ can be expressed as

$$X_{l+1} = \sigma\left(W_l X_l\right) \tag{22}$$

Assume there exists a constant $c_0 > 0$, and $\varepsilon > 0$ such that

$$\varphi_{min}\left(X_1\right) \geq c_0 \tag{23}$$

$$u_l = \gamma_l \varphi_{min}\left(W_l\right) \geq 1 + \varepsilon \tag{24}$$

where $\gamma_l$ is an arbitrary constant.

Then,

$$\varphi_{min}\left(X_{l+1}\right) = \varphi_{min}\left(\sigma\left(W_l X_l\right)\right) \geq \gamma_l \varphi_{min}\left(W_l\right) \varphi_{min}\left(X_l\right) = u_l \varphi_{min}\left(X_l\right) \tag{25}$$

By induction, it follows that

$$\varphi_{min}\left(X_l\right) \geq c_0 \prod_{k=1}^{l-1} u_k \geq c_0 (1 + \varepsilon)^{l-1} \tag{26}$$

Since the Hessian matrix $H = XX^\top$, we have

$$\lambda_{min}\left(H\right) \geq \varphi_{min}(X_l)^2 \geq c_0{}^2 (1 + \varepsilon)^{2(l-1)} \tag{27}$$

As the Hessian matrix $H$ is a symmetric positive definite matrix,

$$\left[H^{-1}\right]_{jj} \leq \|H^{-1}\|_2 = \frac{1}{\lambda_{min}\left(H\right)} \Rightarrow \frac{1}{\left(\left[H^{-1}\right]_{jj}\right)^2} \geq \lambda_{min}(H)^2 \tag{28}$$

Combining the above, we obtain

$$s_{i,j} = \frac{w_{i,j}^2}{\left(\left[H^{-1}\right]_{jj}\right)^2} \geq w_{i,j}^2 \lambda_{min}(H)^2 \geq c_0{}^4 w_{i,j}^2 (1+\varepsilon)^{4(l-1)} \tag{29}$$

Therefore, from equation (29), it follows that the sensitivity scores of the elements in each layer increase exponentially with the layer index $l$.

**Thoerem 3.** Suppose a large language model consists of L layers. As the quantized network goes deeper, the quantization error accumulates progressively with the layer depth, and once a certain threshold is reached, an exponential explosion occurs.

**Proof 3.** Let the full-precision weight matrices of layers $l-1$ and $l$ be denoted by $W_{l-1}, W_l \in R^{k \times n}$, and the quantized weight matrices of layers $l-1$ and $l$ be denoted by $\hat{W}_{l-1}, \hat{W}_l \in R^{k \times n}$. $\sigma$ denotes the activation function of the linear layer.

The activation values at layer $l-1$ in the full-precision model and the quantized model are respectively

$$X_l = \sigma\left(W_{l-1} X_{l-1}\right) \tag{30}$$

$$\hat{X}_l = \sigma\left(\hat{W}_{l-1} \hat{X}_{l-1}\right) \tag{31}$$

The discrepancy of activation values between the full-precision model and the quantized model is denoted as

$$\delta_l = X_l - \hat{X}_l \tag{32}$$

The discrepancy between the full-precision weight matrix and the quantized weight matrix is denoted as

$$E_l = W_l - \hat{W}_l \tag{33}$$

Combining (30), (31), and (32), we obtain

$$\delta_l = \sigma\left(W_{l-1} X_{l-1}\right) - \sigma\left(\hat{W}_{l-1} \hat{X}_{l-1}\right) \tag{34}$$

Substituting the intermediate term $\sigma\left(W_{l-1} \hat{X}_{l-1}\right)$, we obtain

$$\delta_l = \left(\sigma\left(W_{l-1} X_{l-1}\right) - \sigma\left(W_{l-1} \hat{X}_{l-1}\right)\right) + \left(\sigma\left(W_{l-1} \hat{X}_{l-1}\right) - \sigma\left(\hat{W}_{l-1} \hat{X}_{l-1}\right)\right) \tag{35}$$

Let $A_l = \sigma\left(W_{l-1} X_{l-1}\right) - \sigma\left(W_{l-1} \hat{X}_{l-1}\right)$ denoting the propagated error term induced by the quantization of the previous layer.

Let $B_l = \sigma\left(W_{l-1} \hat{X}_{l-1}\right) - \sigma\left(\hat{W}_{l-1} \hat{X}_{l-1}\right)$ representing the sensitivity part associated with the quantization of the current layer.

The Jacobian matrix associated with the activation function $\sigma$ is given by

$$J = \text{diag}(\sigma\left(WX\right)) \tag{36}$$

Applying a first-order Taylor expansion to the activation function $\sigma$, we have

$$\sigma\left(U\right) - \sigma\left(V\right) = J\left(U - V\right) + O\left(||U - V||^2\right) \tag{37}$$

Thus,

$$A_l = J\left(W_{l-1}X_{l-1} - W_{l-1}\hat{X}_{l-1}\right) = JW_{l-1}\delta_{l-1} \tag{38}$$

$$B_l = J\left(W_{l-1}\hat{X}_{l-1} - \hat{W}_{l-1}\hat{X}_{l-1}\right) = JE_{l-1}\hat{X}_{l-1} \tag{39}$$

Therefore,

$$\delta_l = JW_{l-1}\delta_{l-1} + JE_{l-1}\hat{X}_{l-1} \tag{40}$$

We further define the propagation operator and the residual term within the layer as follows:

$$P_{l-1} = JW_{l-1}, \ \xi_{l-1} = JE_{l-1}\hat{X}_{l-1} \tag{41}$$

Hence,

$$\delta_l = P_{l-1}\delta_{l-1} + \xi_{l-1} \tag{42}$$

Expanding recursively with respect to (42), we have:

$$\delta_l = P_{l-1}\delta_{l-1} + \xi_{l-1} = P_{l-1}\left(P_{l-2}\delta_{l-2} + \xi_{l-2}\right) + \xi_{l-1} = P_{l-1}P_{l-2}\delta_{l-2} + P_{l-1}\xi_{l-2} + \xi_{l-1} \tag{43}$$

By recursively expanding across layers, we finally obtain

$$\delta_l = \sum_{r=1}^{l-1}\left(\prod_{s=r+1}^{l-1} P_s\right)\xi_r \tag{44}$$

We can further reorganize the above expression into a more compact form:

$$\delta_l = \sum_{r=1}^{l-1} G_{r\to l-1}\delta_r, \ G_{r\to l-1} = \prod_{s=r+1}^{l-1} P_s = \prod_{s=r+1}^{l-1} JW_s \tag{45}$$

Here, $G_{r\to l-1}$ can be interpreted as the error-propagation operator that maps the quantization-induced perturbation at layer $r$ to layer $l$. According to equation (45), during quantization, the quantization error introduced in earlier layers continuously accumulates and propagates forward. Moreover, the full-precision weight norms of different layers directly influence the magnitude of this propagation. This further demonstrates the importance of leveraging full-precision weight norms as a relevance indicator when evaluating the significance of weight elements.

Let $a_l = ||\delta_l||_F$. Assume that the quantization error of the activation value at each layer is bounded and non-degenerate, i.e.,

$$C_{\min} \le ||\widehat{X_l}||_F \le C_{\max}, \alpha \le ||E_l||_2 \le \beta, l = 1, ..., L - 1 \tag{46}$$

and assume that each weight matrix satisfies

$$\gamma_l\sigma\left(W_l\right) \ge 1 + \varepsilon, \varepsilon > 0 \tag{47}$$

where $\gamma_l$ is an arbitrary constant.

Then

$$\begin{aligned}
a_{l+1} &= ||\sigma\left(W_lX_l\right) - \sigma\left(W_l + E_l\right)\widehat{X_l}||_F \\
&\ge \gamma_l||W_l\delta_l - E_l\widehat{X_l}||_F \\
&\ge \gamma_l\left(\sigma\left(W_l\right)a_l - ||E_l||_2||\widehat{X_l}||_F\right) \\
&\ge (1 + \varepsilon)a_l - \gamma_{max}\beta C_{max}
\end{aligned} \tag{48}$$

Let $b = \gamma_{max}\beta C_{max}$.

By solving the recursive inequality (48), we obtain

$$a_l \geq (1+\varepsilon)^{l-1}\left(a_1 - \frac{b}{\varepsilon}\right) + \frac{b}{\varepsilon} \tag{49}$$

If there exists some index $l_0$ such that $a_{l_0} > \frac{b}{\varepsilon}$, then for any $k > 0$, we have

$$a_{l_0+k} \geq c(1+\varepsilon)^k, c = a_{l_0} - \frac{b}{\varepsilon} > 0 \tag{50}$$

Thus,

$$a_L \geq c(1+\varepsilon)^{L-l_0} = \Omega\left((1+\varepsilon)^L\right) \tag{51}$$

As given by equation (51), in the quantization process, the absolute value of the quantization error will increase with the layer depth. Once $a_l$ exceeds the threshold $\frac{b}{\varepsilon}$, the quantization error will exhibit an exponential blow-up.

## C    THE THEORY ANALYSIS AND EMPIRICAL DISTRIBUTIONS OF ATTRIBUTE OUTLIERS

For the quantization task of large language models, different weight elements have different impacts on model performance after quantization. Weight elements that have a relatively large impact on model performance are usually regarded as outliers. However, many previous methods only defined outliers from a single numerical perspective(Shao et al., 2023; Lin et al., 2024a). In this section, we attempt to analyze and reveal the influence of the Hessian properties corresponding to weight elements on the quantization performance of the model, and propose the concept of "attribute outliers" which are elements with normal values but having a significant impact on model performance.

In the field of model compression, early pruning methods(LeCun et al., 1989) approximate the loss error $\varepsilon$ caused by numerical perturbations $w$ to weight elements through network convergence $\frac{\partial \mathcal{L}}{\partial w}$ and Taylor series:

$$\varepsilon = \left(\frac{\partial \mathcal{L}}{\partial w}\right)^{\top} w + \frac{1}{2}\frac{\partial^2 \mathcal{L}}{\partial w^2}w^2 + O(w^3) \approx \frac{1}{2}\frac{\partial^2 \mathcal{L}}{\partial w^2}w^2 \tag{52}$$

This formula shows that not only the value of weight elements affects model performance, but also the second-order derivative of weight elements with respect to the loss error can influence model performance. Due to the large number of parameters in large language models, directly calculating $\frac{\partial^2 \mathcal{L}}{\partial w^2}$ is difficult. The OBC(Frantar & Alistarh, 2022) method attempts to transform $\frac{\partial^2 \mathcal{L}}{\partial w^2}$ into a computationally-friendly expression to serve model lightweighting. Specifically, it first constructs an objective function:

$$\arg\min\left\|WX - \hat{W}X\right\| \tag{53}$$

where W, $\hat{W}$, and X represent the weight matrix, quantized weight matrix, and input matrix of the linear layer, respectively.

Then, based on this objective function, by differentiating with respect to W, it obtains an approximate solution of $\frac{\partial^2 \mathcal{L}}{\partial w^2}$, namely the Hessian matrix:

$$H = \frac{\partial^2 \mathcal{L}}{\partial w^2} \approx 2XX^{\top} \tag{54}$$

Assuming that the currently quantized element is the $q$-th dimensional element $w_q$, OBC(Frantar & Alistarh, 2022) attempts to adjust $\Delta w$ for other unquantized elements to minimize the quantization

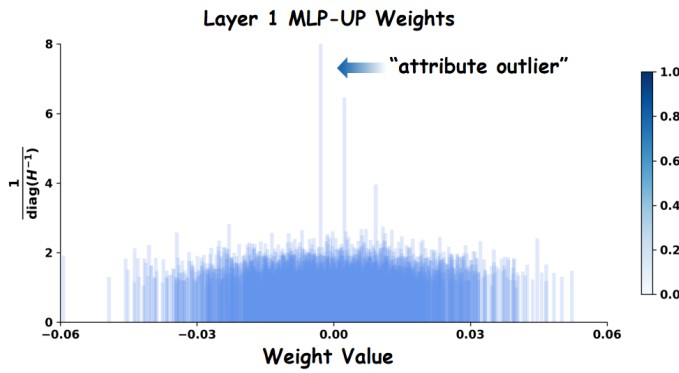

Figure 5: The distributions of Hessian attributes and numerical magnitude in LLaMA2-7B.

loss. It combines the constraint condition $e_q^\top \Delta w - (\text{quant}(w_q) - w_q) = 0.$ with the optimization function to obtain the Lagrangian equation:

$$\varepsilon = L(\Delta w, \lambda) = \frac{1}{2}\Delta w^\top H \Delta w + \lambda \left(e_q^\top \Delta w - (\text{quant}(w_q) - w_q)\right) \tag{55}$$

The solution method based on the Lagrangian equation can yield the loss error $\varepsilon$:

$$\varepsilon = \frac{1}{2} \frac{(\text{quant}(w_q) - w_q)^2}{H_{q,q}^{-1}} \tag{56}$$

We can obtain:

$$\varepsilon \propto (\text{quant}(w_q) - w_q)^2, \varepsilon \propto \frac{1}{H_{q,q}^{-1}} \tag{57}$$

This conclusion further illustrates that in the quantization process, the dynamic quantization error and the static Hessian attribute jointly determine their impact on the overall performance loss.

As shown in Fig.5, we further explore the distributions of Hessian attributes and numerical magnitude of weight elements in LLaMA-2-7B and demonstrate the presence of "attribute outliers"—weights that are sensitive to quantization but not numerically large.

## D    DETAILED RESULTS OF FLEXIBLELLM ON ZERO-SHOT EVALUATION TASK

In this section, as shown in Table 6, we present a comprehensive and detailed presentation of the performance of FlexibleLLM across five specific zero-shot tasks.

## E    DESIGN AND IMPLEMENTATION DETIALS OF NOR MODULE

In recent years, random Hadamard transforms have been widely used in quantization to suppress outliers in weightsAshkboos et al. (2024); Liu et al. (2024). However, existing Hadamard matrices are predominantly constructed through recursive procedures. Each recursive layer introduces new sign-composition patterns and recursive paths, yielding basis vectors with markedly different distribution characteristics. Moreover, the inherent randomness and instability of these basis vectors disrupt the relative dimensional structure of the weight matrix. To address this issue, our NOR module applies fine-grained denoising and reorganization to the randomly generated Hadamard matrix, aiming to construct a smoother and more structurally consistent transform matrix. The NOR consists of three main components: feature engineering, local K-means clustering, and two-level sorting.

1) **Feature Engineering**: In this stage, we construct a 7-dimensional feature vector for each basis vector to comprehensively capture its structural characteristics. The features include:

*Global Pattern Complexity: The total number of sign transitions between adjacent elements.*

Table 6: Zero-shot performance of FlexibleLLM on five common tasks.

| model | method | Bits | PIQA | HellaSwag | ARC-c | ARC-e | Winogrande | Avg. |
|---|---|---|---|---|---|---|---|---|
| LLaMA-7B | FlexibleLLM | 2.0 | 67.68 | 51.01 | 48.48 | 28.16 | 59.75 | 51.02 |
| | FlexibleLLM | 2.1 | 71.16 | 58.17 | 54.76 | 31.31 | 61.56 | 55.39 |
| | FlexibleLLM | 3.0 | 78.35 | 72.88 | 70.71 | 43.00 | 68.35 | 66.66 |
| LLaMA-13B | FlexibleLLM | 2.0 | 73.83 | 63.15 | 62.08 | 36.35 | 66.38 | 60.36 |
| | FlexibleLLM | 2.1 | 74.43 | 66.51 | 67.34 | 37.63 | 67.64 | 62.71 |
| | FlexibleLLM | 3.0 | 79.05 | 76.94 | 74.43 | 47.10 | 70.88 | 69.68 |
| LLaMA-30B | FlexibleLLM | 2.0 | 76.44 | 70.05 | 70.16 | 42.75 | 70.56 | 65.99 |
| | FlexibleLLM | 2.1 | 77.42 | 72.25 | 70.41 | 42.83 | 71.35 | 66.85 |
| | FlexibleLLM | 3.0 | 80.14 | 80.79 | 78.28 | 52.13 | 74.90 | 73.25 |
| LLaMA-2-7B | FlexibleLLM | 2.0 | 68.77 | 50.85 | 53.87 | 28.84 | 61.17 | 52.70 |
| | FlexibleLLM | 2.1 | 71.87 | 58.55 | 58.88 | 31.57 | 62.75 | 56.72 |
| | FlexibleLLM | 3.0 | 77.91 | 72.60 | 72.14 | 43.52 | 67.48 | 66.73 |
| LLaMA-2-13B | FlexibleLLM | 2.0 | 71.98 | 60.96 | 63.76 | 34.22 | 64.48 | 59.08 |
| | FlexibleLLM | 2.1 | 73.34 | 64.06 | 65.45 | 36.60 | 65.11 | 60.91 |
| | FlexibleLLM | 3.0 | 78.62 | 76.73 | 74.24 | 46.25 | 70.17 | 69.20 |
| LLaMA-3-8B | FlexibleLLM | 2.0 | 69.31 | 51.86 | 59.92 | 32.76 | 60.77 | 54.92 |
| | FlexibleLLM | 2.1 | 69.91 | 54.24 | 59.22 | 34.04 | 63.06 | 56.09 |
| | FlexibleLLM | 3.0 | 78.29 | 75.40 | 74.71 | 48.21 | 71.51 | 69.62 |
| LLaMA-3-70B | FlexibleLLM | 2.0 | 72.52 | 56.53 | 66.58 | 39.93 | 71.03 | 61.32 |
| | FlexibleLLM | 2.1 | 77.20 | 67.96 | 74.33 | 45.65 | 72.61 | 67.55 |
| | FlexibleLLM | 3.0 | 83.13 | 82.53 | 83.25 | 59.90 | 78.14 | 77.39 |

*Maximum Coherence Length: The length of the longest consecutive same-sign segment.*

*Average Coherence Length: The mean length of all same-sign segments.*

*Spatial Coherence: The variance of pattern complexity across several equal-length subsegments.*

*Sign Balance: The proportion of positive elements.*

*Prefix Local Complexity: The pattern complexity within the first quarter of the vector.*

*Suffix Local Complexity: The pattern complexity within the last quarter of the vector.*

2) **Local K-means Clustering**: Based on the feature vectors constructed above, all basis vectors are grouped using K-means clustering. The algorithm partitions them into different clusters by iteratively minimizing within-cluster variance, ensuring that basis vectors with similar structural patterns are automatically grouped together.

3) **Two-level Sorting**: In this stage, we generate the final ordering of basis vectors using a two-level strategy that proceeds from local to global organization. We first perform fine-grained optimization within each cluster by sorting the basis vectors in ascending order according to the Euclidean distance between their feature vectors and the corresponding cluster center. Next, at the inter-cluster level, we apply coarse-grained sorting based on the average pattern complexity of each cluster. The original Hadamard matrix is then reorganized according to the resulting two-stage ordering.

## F  DESIGN AND IMPLEMENTATION DETIALS OF LFD MODULE

### F.1  LAYER-WISE ADAPTIVE COMPENSATION STRATEGY

For LLMs, Our analyses in the Appendix and the Introduction provide both empirical and theoretical evidence that variant layers show differences in static sensitivity scores and dynamic quantization errors. These differences arise because some layers are mainly responsible for constructing low-level representations and maintaining model robustness, while others are directly involved in generating and are therefore highly sensitive to quantization. This indicates that applying a uniform compensation strength across all layers during quantization may harm both robustness and accuracy. Ac-

cordingly, we introduce a layer-wise adaptive compensation strategy that infers the functional role of each layer from its input–output similarity and adjusts the strength of quantization compensation in a dynamic and flexible manner.

For the $\ell$-th layer, we perform a forward pass on the full-precision model to obtain its input representation $X_\ell$ and output representation $Y_\ell$. We then compute the cosine similarity between them on a per-token basis and take the average.

$$S_\ell = \frac{1}{NL} \sum_{n,l}^{N,L} \frac{X_\ell^{(n,l)} \cdot Y_\ell^{(n,l)}}{||X_\ell^{(n,l)}||||Y_\ell^{(n,l)}||} \tag{58}$$

Where $N$ and $L$ denote the number of samples and the sequence length, respectively. This metric directly reflects the functional role of each layer. A higher similarity indicates that the layer primarily contributes to low-level representation construction and thus plays a role in maintaining model robustness. In contrast, a lower similarity suggests that the layer performs substantial nonlinear transformations and information reorganization, making it highly sensitive to quantization.

After obtaining similarity scores for all layers, we first normalize these scores to eliminate the effect of absolute magnitudes and to capture the relative differences across layers.

$$S_\ell = \frac{S_\ell - S_{min}}{S_{max} - S_{min}} \tag{59}$$

Where $S_\ell$ denotes the input–output similarity score of the $\ell$-th layer. $S_{max}$ and $S_{min}$ represent the maximum and minimum input–output similarity scores across all layers, respectively.

Then, we establish an inverse mapping between the similarity scores and the compensation coefficients $\alpha_\ell$. Layers with simpler functional roles receive weaker compensation to preserve their inherent robustness, whereas those with more complex functions receive stronger compensation to ensure that critical generation and reasoning capabilities are accurately preserved.

$$\alpha_\ell = \alpha_{max} - S_\ell \times (\alpha_{max} - \alpha_{min}) \tag{60}$$

By combining equation (10), we obtain the final form of total quantization error compensation term:

$$\delta_i^{(w)} = \frac{quant(w_i) - w_i}{H_{i,i}^{-1}} H_{:,i}^{-1}, \delta_i^{(a)} = W_\ell^{row}(X_\ell - \widehat{X_\ell})\widehat{X_\ell}^\top H_{-i}^{-1}$$
$$\delta_i = \delta_i^{(w)} + \alpha_\ell \delta_i^{(a)} \tag{61}$$

This layer-aware calibration and compensation strategy avoids the dilemma of uniform compensation, which can lead to either excessive correction or insufficient adjustment, and thereby achieves a balanced trade-off between model robustness and accuracy.

### F.2 EFFICIENT GROUPED STORAGE MECHANISM AND PIPELINE-COORDINATED SLIDING-WINDOW MECHANISM

Using full-precision activations for quantization-error compensation requires accessing the corresponding activation tensors for each Transformer layer. Storing all of these activations of a layer on the GPU at once is prohibitively expensive for ordinary users due to the large memory footprint. An intuitive alternative is to move them to the CPU and send them back to the GPU on demand, but this leads to a substantial increase in runtime because of frequent CPU–GPU data transfers. Each Transformer layer consists of seven linear layers, and due to their structural properties, several of these linear layers share identical input activations. As shown in Table 7, this fact motivates us group the seven linear layers into four distinct groups, where each group shares the same full-precision activations for calibration. This grouping reduces the activation-storage overhead to some extent.

Additionally, since current quantization algorithms process groups sequentially, we further observe two key properties: 1) **Temporal separation of access patterns**: different groups require activation

Table 7: The details of the grouping scheme applied to the various linear layers. g1, g2, g3, g4 respectively denote the groups to which different linear layers belong.

| Linear layer | self_attn.q_proj | self_attn.k_proj | self_attn.v_proj | self_attn.o_proj | mlp.up_proj | mlp.gate | mlp.down |
|---|---|---|---|---|---|---|---|
| Group domain | g1 | g1 | g1 | g2 | g3 | g3 | g4 |

access at clearly different time points during quantization. 2) **A deterministic gap between computation and transfer**: the time required for quantization computation is significantly longer than the corresponding CPU–GPU transfer time. These observations motivate the design of a pipeline-coordinated sliding-window mechanism. The mechanism keeps only a limited number of activation groups on the GPU and asynchronously prefetches the next group during the quantization of the current one. Because the transfer time is fully hidden behind computation, this approach achieves nearly negligible time overhead while greatly reducing GPU memory usage. Specifically, during the quantization phase, we maintain a sliding window of size two, ensuring that at most two activation groups reside on the GPU at any moment. The window simultaneously covers the group currently being quantized and the next group in the processing order. When quantizing the current group, the algorithm initiates an asynchronous prefetch of the next group using an independent CUDA stream, enabling full overlap between data transfer and computation. Since quantization time far exceeds transfer time, the prefetched group is always ready by the time the window advances, ensuring seamless switching. As the window slides forward, processed groups are immediately evicted from GPU memory and replaced by newly prefetched ones, forming a pipeline-style data flow. This design fully hides data-transfer latency and tightly constrains the GPU memory cost of activations to that of only two groups.

# G  FUTURE WORK

For FlexibleLLM, while its LFD module leverages a layer-wise adaptive compensation mechanism in appendix F.1 to account for the heterogeneity in LLMs, this compensation framework based on input-output similarity scores currently only captures the global coarse-grained discrepancies across layers during processing and fails to further accommodate the local fine-grained differences among elements within each layer. In future work, we will explore the possibility of addressing quantization errors caused by outliers through the integration of both coarse-grained and fine-grained weight information in LFD-PRO. Specifically, during the quantization process of each layer, we will not only utilize the global layer-level input-output similarity, but also incorporate the local element-level static sensitivity scores to the whole quantization process. The overall alignment optimization objective for LFD-PRO is as follows:

$$\underbrace{\arg\min_{\widehat{W}_\ell} \left\| W_\ell X_\ell - \widehat{W}_\ell \widehat{X}_\ell \right\|_2^2}_{LFD} \implies \underbrace{\arg\min_{\widehat{W}_\ell} \left\| (Q \odot W_\ell) X_\ell - (Q \odot \widehat{W}_\ell) \widehat{X}_\ell \right\|_2^2}_{LFD-PRO} \tag{62}$$

where $W_\ell$ and $X_\ell$ represent the full-precision weight matrix and the input activation of the $\ell$-th layer. $\widehat{W}_\ell$ and $\widehat{X}_\ell$ represent the quantized weight matrix and the input activation of the $\ell$-th layer. $Q$ denotes the static sensitivity score matrix corresponding to the weight matrix. We can further derive the quantization error compensation term $\delta_i$ using the Lagrangian function:

$$\delta_i = \frac{q_i(quant(w_i) - w_i)}{H_{i,i}^{-1}} H_{:,i}^{-1} + \alpha_\ell (Q \odot W_\ell^{row})(X_\ell - \widehat{X}_\ell)\widehat{X}_\ell^\top H_{-i}^{-1} \tag{63}$$

$\alpha_\ell$ denotes the layer-aware factor for the $\ell$-th layer, which serves to prevent the generalization and robustness of the quantized model. We can then obtain:

$$\delta_i = \underbrace{\delta_i^{(w)} + \alpha_\ell \delta_i^{(a)}}_{coarse-grained compensation\ driven\ by\ layer-leval\ similarity} \tag{64}$$

$$\underbrace{\delta_i^{(w)} = \frac{q_i(quant(w_i) - w_i)}{H_{i,i}^{-1}} H_{:,i}^{-1}, \delta_i^{(a)} = (Q \odot W_\ell^{row})(X_\ell - \widehat{X_\ell})\widehat{X_\ell}^\top H_{-i}^{-1}}_{fine-grained\ driven\ by\ element-level\ importence\ score} \tag{65}$$

## G.1 ADDITIONAL ABLATION STUDY

As shown in Table 8 , we conducted ablation experiments to directly compare the Quarot and the NOR module, the results reveal that the NOR module achieves substantial performance improvements in model quantization tasks. Additionally, we also compare the performance directly between GPTQ and LFD. The experimental results on Llama1-7B and Llama1-13B presented in Table 9 validate the role of layer-wise adaptive adjustment for performance improvement. We also comprehensively compare the computation time and GPU memory requirements of LFD, GPTAQ and the variant of GPTAQ (ALL_TO_CPU) which transfers the all activations of a layer to CPU during quantization in Table 10. The results reveal that LFD can complete quantization with much fewer valuable GPU memory resources and comparable quantization time compared with GPTAQ.

Table 8: Comparison of Quarot and NOR on Llama1-7B and Llama1-13B.

| Bit | Model | Method | lambada_openai | lambada_standard | MathQA | MMLU_avg | WikiText2 perplexity |
|---|---|---|---|---|---|---|---|
| 2 bit | Llama1-7B | Quarot | 12.74 | 11.68 | 20.46 | **23.76** | 18.07 |
| 2 bit | Llama1-7B | NOR | **19.86** | **17.26** | **22.65** | 23.18 | **15.19** |
| 2 bit | Llama1-13B | Quarot | 13.56 | 10.11 | 21.40 | 23.94 | 16.15 |
| 2 bit | Llama1-13B | NOR | **21.02** | **11.57** | **23.26** | **24.40** | **12.53** |

Table 9: Comparison of GPTAQ and LFD on Llama1-7B and Llama1-13B.

| Bit | Model | Method | lambada_openai | lambada_standard | MathQA | MMLU_avg | WikiText2 perplexity |
|---|---|---|---|---|---|---|---|
| 2 bit | Llama1-7B | GPTAQ | 12.15 | 11.42 | 21.26 | 22.76 | 12.91 |
| 2 bit | Llama1-7B | LFD | **14.92** | **13.06** | **22.02** | **23.05** | **12.12** |
| 2 bit | Llama1-13B | GPTAQ | 16.62 | 16.84 | 21.65 | 23.08 | 10.81 |
| 2 bit | Llama1-13B | LFD | **17.79** | **18.37** | **22.38** | **23.75** | **10.33** |

Table 10: Comparison of GPU memory and quantization time per layer of Llama3-8B and Llama3-70B.

| Method | Llama3-8B Time | Llama3-8B Memory | Llama3-70B Time | Llama3-70B Memory |
|---|---|---|---|---|
| GPTAQ | **45.9s** | 25.86GB | **140.2s** | 63.13GB |
| GPTAQ (ALL_TO_CPU) | 55.58s | 12.38GB | 170.4s | 35.28GB |
| LFD | 46.5s | **10.54GB** | 141.4s | **30.85GB** |

