# OpenReview forum: "FlexibleLLM: Making Low-Bit Quantization for Large Language Models More Flexible and Efficient"
_ICLR.cc/2026/Conference — Submitted to ICLR 2026_

### Official Review · Reviewer_zVui · 2025-10-18

**Soundness:** 2
**Presentation:** 1
**Contribution:** 1
**Rating:** 2
**Confidence:** 5

**Summary:**

This work presents FlexibleLLM, a weight-only post-training quantization method. At its core, three components are introduced: (1) A mixed-precision within each layer, (2) an outlier reduction mechanism with Hadamard transform, (3) asymmetric calibration. Unfortunately, I haven't found any of these to appear novel, given that they are already proposed in other works.

**Strengths:**

The name of each submodule looks fancy.

**Weaknesses:**

My major concern for this work is its novelty.

+ For SBGBS, the sailency metric has been proposed in SliM-LLM [1], in which the Hessian diagonal values and the weight norm are used. In this work, the authors used the weight norm, not the quantization error norm, which makes less sense given that it is not pruning. In the meantime, this metric is not ACCURATE since the weights are updated after every iteration. This problem has been discussed in the SparseGPT Algorithm, but I do not see any deep discussion in this paper.

+ For NOR, this is the same as QuaRot. And more ironically, even the code implementation is the same as QuaRot without citing them in Section 3.2.1. Apparently, the authors know what they are doing.

+ For LFD, the core idea and methodology are exactly the same as GPTAQ [2]. Another proof is that they used the code from GPTAQ by slightly modifying the function name. Yet they did not compare the GPTAQ in the experiments.

I would reconsider my rating if the authors could acknowledge they used the implementation of QuaRot and GPTAQ, clarify the difference, if any, and provide the file `flexiblellm_utils.py` for a detailed evaluation.

## Minor weakness

+ The notation system in this work is very chaotic. Hessian is represented with both $H$ and $\mathrm{H}$.

+ The evaluation is primarily about perplexity, which is less trustworthy in the current LLM community. I would like to see MMLU and reasoning task accuracy.

## Reference

[1] SliM-LLM: Salience-Driven Mixed-Precision Quantization for Large Language Models, ICML 2025.

[2] GPTAQ: Efficient Finetuning-Free Quantization for Asymmetric Calibration, ICML 2025.

**Questions:**

None.

---

> ### Author Response · Authors · 2025-12-04
> **Author Responses (1/n)**
>
> > **W1(1)**: SBGBS module used the weight norm, not the quantization error norm.
>
> We sincerely appreciate the reviewer’s valuable feedback. We would like to clarify that the SBGBS module does not rely solely on a single static weight-norm metric.
> - Instead, it jointly leverages the **static** weight norm, the diagonal Hessian information, and the **dynamic** quantization-error norm to assign higher bit-widths to regions where outliers are more densely concentrated **(please refer to Sec. 3.1 and Figure 2 of thr paper)**. The greedy bit-width allocation strategy further enhances the interpretability of our method.
> - Moreover, in the ablation studies, we directly compared SBGBS with recent related **static** approaches such as SLIM-LLM, and we systematically summarized two key factors that account for the superior performance of SBGBS **(please refer to Sec. 4.3 of the paper)**.
>
> > **W1(2)**: Weight norm from original full-precision model may not accurate compared to iteratively updated weight norms during quantization.
>
> We thank the reviewer for raising this insightful point. Indeed, the iterative update strategy in SparseGPT may appear reasonable from an intuitive perspective. However, this strategy involves a substantial risk of collapse under the low-bit quantization setting considered in our work.
> - In essence, SparseGPT depends on a partially quantized intermediate model to estimate the element-wise importance of the weight matrix. In low-bit scenarios, such intermediate models typically suffer from severe performance degradation or even complete collapse after most quantization steps. Consequently, the iteratively updated weights are unreliable and heavily contaminated by noise.
>
> - Moreover, the above-mentioned view that information from quantized intermediate model (including weights, activations, and gradients) is unreliable has also been raised in recent literature. For instance, a recent work[1] on MoE quantization points out that quantized intermediate activations can give rise to an “expert shift” issue.
>
> - Additionally, during the rebuttal phase we further included a theoretical analysis in the **Thoerem 3 of Appendix B**, showing that the propagation strength of the quantization error at each layer is positively correlated with the norm of that layer’s full-precision weights. This result underscores the importance of the original full-precision weight norms.
>
> - To sum up, our SBGBS module jointly utilizes **static** sensitivity scores obtained from the full-precision model with the **dynamic** quantization-error norm. This design preserves the dynamic characteristics of the quantization process while effectively avoiding the collapse risk associated with importance estimation from degraded intermediate models.
>
> [1] Chen, Yuanteng, et al. "EAC-MoE: Expert-selection aware compressor for mixture-of-experts large language models." Proceedings of the 63rd Annual Meeting of the Association for Computational Linguistics (Volume 1: Long Papers). 2025.
>
>
> > **W2(2),W3(2)**: The original version of code is overlapped with some existing codebases.
>
> Firstly, regarding the code-related concerns raised in both Weakness 2 and 3, we sincerely apologize for the confusion caused. The overlap between our submitted code and other implementations arises from our institutional restrictions that we are currently prohibited from uploading the original project source code to public platforms. Consequently, the code provided in the anonymous link was a simplified twin version based on the open-source GPTAQ repository, with the purpose of allowing reviewers to understand the overall workflow of our algorithm.
>
> In addition, we have noticed discrepancies between the code previously shared in the anonymous link and the final version of our twin implementation. This inconsistency may have caused confusion and given the impression that our NOR and Quarot implementations are identical. We have now uploaded a complete and consistent version to the anonymous repository. We further promise that, once our institutional restrictions are lifted, we will immediately release the original project source code(including preprocessing, quantization, unpacking, packing, kernel implementations, and all related modules) on GitHub or other public platforms.

---

> ### Author Response · Authors · 2025-12-04
> **Author Responses (2/n)**
>
> > **W2(1)**: Difference between NOR and Quarot.
>
> We appreciate the reviewer’s valuable and insightful feedback. Although both Quarot and the NOR module involve the use of Hadamard matrices, it should be clarified that our method differs essentially from the Quarot method:
> - Specifically, our method accounts for the noise issue of randomly generated Hadamard matrices and performs denoising optimization based on feature engineering, local K-means clustering, and two-level sorting, effectively avoiding the negative impact of random Hadamard matrices on the relative distribution of weight elements **(more details in Appendix E of the paper)**.
> - In contrast, although the random generation of Hadamard matrix can alter and even destroy the relative structure of the original weights, Quarot does not provide any analysis or remedy for this potential degradation.
>
>
> Additionally, during the rebuttal stage, we conducted ablation experiments to directly compare the Quarot and the NOR module. The results are shown below:
>
> |  Bit  |   Model    | Method | lambada_openai | lambada_standard | mathqa | MMLU_avg | WikiText2 perplexity |
> |:-----:|:----------:|:------:|:--------------:|:----------------:|:------:|:--------:|:--------------------:|
> | 2 bit | Llama1-7B  | Quarot |     12.74      |      11.68       | 20.46  |  **23.76**   |        18.07         |
> | 2 bit | Llama1-7B  |  NOR   |     **19.86**      |      **17.26**       | **22.65**  |  23.18   |        **15.19**         |
> | 2 bit | Llama1-13B  | Quarot |      13.56      |       10.11       | 21.40  |  23.94   |        16.15         |
> | 2 bit | Llama1-13B  |  NOR   |     **21.02**      |      **11.57**       | **23.26**  |  **24.40**   |        **12.53**         |
>
>
> The results reveal that the NOR module achieves substantial performance improvements in model quantization tasks.

---

> ### Author Response · Authors · 2025-12-04
> **Author Responses (3/n)**
>
> > **W3(1)**: Difference between LFD and GPTAQ.
>
> We appreciate the reviewer’s attention to this issue. First, before illustrating the differences between the LFD module and GPTAQ, we acknowledge that GPTAQ is an extraordinary and outstanding work. It has made tremendous contributions to the development of the quantization field with only 20 lines of code, and both it and the LFD module utilize full-precision activations for additional error compensation based on the GPTQ method. Next, we will detail the differences between GPTAQ and the LFD module from three dimensions including theoretical analysis, method design and resources optimization:
>
> - **Theoretical analysis**
>   - **Our work is among the few that directly propose a multi-scale and comprehensive analysis method targeting the outlier issue that limits quantization performance.** Specifically, we first systematically characterizes the spatial distribution patterns and variation trends of outliers and their introduced quantization errors from the perspectives of numerical magnitude and intrinsic attributes, thereby providing new potential inspiration for future quantization methods.
>
> - **Method design**
>   - Quantization error compensation methods with full-precision activations will ultimately generate a compensation term. GPTAQ uses a unified compensation coefficient to constrain the error compensation term. However, since different layers in LLMs play different roles, enforcing a uniform correction strength across all layers can be suboptimal and may inadvertently undermine the robustness of intermediate and shallow layers.
>   - Unlike strategies with fixed compensation coefficients, the LFD module adaptively adjusts the intensity of this term at the layer level according to each layer’s importance score, better matches the actual functional role of each layer rather than treating all layers identically(**More details in Appendix F**).
>   - The experimental results on Llama1-7B and Llama1-13B presented in **Table 1** validate the role of layer-wise adaptive adjustment for performance improvement.
>   - Meanwhile, we also discuss the possibility to further upgrade the LFD by jointly leveraging coarse-grained layer-wise importance information and fine-grained element-level static sensitivity information to more comprehensively  in the future work section of **Appendix G**. We also hope that it can provide inspiration for other researchers in the community.
>
>  **Table 1: Comparison of GPTAQ and LFD on Llama1-7B and Llama1-13B**
> |  Bit  |   Model    | Method | lambada_openai | lambada_standard | mathqa | MMLU_avg | WikiText2 perplexity |
> |:-----:|:----------:|:------:|:--------------:|:----------------:|:------:|:--------:|:--------------------:|
> | 2 bit | Llama1-7B  | GPTAQ |      12.15      |       11.42       | 21.26  |  22.76   |        12.91         |
> | 2 bit | Llama1-7B  |  LFD   |     **14.92**      |      **13.06**       | **22.02**  |  **23.05**   |        **12.12**         |
> | 2 bit | Llama1-13B  | GPTAQ |     16.62      |      16.84       | 21.65  |  23.08   |        10.81         |
> | 2 bit | Llama1-13B  |  LFD   |     **17.79**      |      **18.37**       | **22.38**  |  **23.75**   |        **10.33**         |

---

> ### Author Response · Authors · 2025-12-04
> **Author Responses (4/n)**
>
> - **Resources optimization**
>   - Full-precision activations require substantial memory overhead. Although GPTAQ try to reduce the GPU memory usage by only storing the activations of one layer duing quantization, it will still consume a lot of GPU memory. A straightforward alternative is to offload them to the CPU and recall them to the GPU on demand, but frequent CPU–GPU data transfers significantly increase runtime.
>   - Unlike the above strategies, our LFD module had effectively reduced the memory requirement during error compensation primarily through efficient grouped storage mechanism and pipeline-coordinated sliding-window mechanism(**More details in Appendix F**).
>   - As detailed in **Table 2**, we comprehensively compare the computation time and GPU memory requirements of LFD, GPTAQ and the variant of GPTAQ(ALL_TO_CPU) which transfers the all activations of a layer to CPU during quantization:
>
>  **Table 2: Comparison of GPU memory and quantization time per layer of Llama3-8B and Llama3-70B**
> | Method     | Llama3-8B Time | Llama3-8B Memory      | Llama3-70B Time | Llama3-70B Memory        |
> |------------|----------------|------------------------|-----------------|---------------------------|
> | GPTAQ      | **45.9s**          | 25.86GB                | **140.2s**          | 63.13GB                   |
> | GPTAQ(ALL_TO_CPU) | 55.58s         | 12.38GB                | 170.4s          | 35.28GB                   |
> | LFD        | 46.5s          | **10.54GB**                | 141.4s          | **30.85GB**                   |
>
> The results reveal that LFD can complete quantization with much fewer valuable GPU memory resources and comparable quantization time compared with GPTAQ. It is worth noting that the LFD requires fewer GPU memory resources and time overhead than GPTAQ(ALL_TO_CPU) during quantization. These results further demonstrate the efficiency of grouped storage mechanism and pipeline-coordinated sliding-window mechanism.
>
>
>
>
> > **W4**: The notation system is chaotic.
>
> We thank the reviewer for the valuable comments. We have carefully re-examined all notation used throughout the manuscript and have made comprehensive and detailed revisions accordingly.
>
> > **W5**: Few evaluation on reasoning tasks.
>
> We appreciate the reviewer’s thorough and detailed comments. To address the raised concerns,  we conducted extensive evaluations on a wider range of reasoning benchmarks, including **MMLU**, **MathQA**, and **LAMBADA**：
>
> |  Bit  |   Model    |   method    | lambada_openai | lambada_standard | mathqa | MMLU_avg | WikiText2 perplexity |
> |:-----:|:----------:|:-----------:|:--------------:|:----------------:|:------:|:--------:|:--------------------:|
> | 2 bit | Llama2-7B  |   GPTQ      |      0.00      |       0.00       | 21.12  |  21.89   |        60.45         |
> | 2 bit | Llama2-7B  | OmniQuant   |     4.02      |      1.78       | 23.74  |  23.71   |        11.06         |
> | 2 bit | Llama2-7B  | AffineQuant |     6.57      |      3.43       | 22.68  |  23.67   |        10.87         |
> | 2 bit | Llama2-7B  |    **Ours**     |     **46.56**      |      **38.73**       | **24.08**  |  **23.93**   |         **8.70**         |
> | 2 bit | Llama1-7B  |   GPTQ      |      0.00      |       0.00       | 21.97  |  21.56   |       152.31         |
> | 2 bit | Llama1-7B  | OmniQuant   |     23.89      |      15.78       | 23.32  |  23.45   |         9.71         |
> | 2 bit | Llama1-7B  | AffineQuant |     22.35      |      15.68       | 22.17  |  **23.51**   |        13.51         |
> | 2 bit | Llama1-7B  |    **Ours**     |     **38.02**      |      **29.14**       | **23.94**  |  23.48   |         **8.82**         |
>
> The experimental results demonstrate that FlexibleLLM consistently maintains its performance advantage over existing methods on these reasoning tasks.

---

### Official Review · Reviewer_YATu · 2025-10-30

**Soundness:** 3
**Presentation:** 3
**Contribution:** 3
**Rating:** 6
**Confidence:** 4

**Summary:**

This paper tackles the challenge that existing post‑training quantization (PTQ) methods fail to adequately handle both “discrete outliers” and “attribute outliers,” and tend to ignore error accumulation across Transformer layers. To address this, the authors propose FlexibleLLM, a finetuning‑free, weight‑only PTQ framework, comprised of three key modules: SBGBS (self‑adaptive block‑level greedy bit search) for flexible fractional bit allocation, DOSA (discrete outlier suppression & awareness) combining Hadamard transforms and Hessian signals, and LFD (layer‑level feedback & denoising) to counter cross‑layer error propagation. Experimental results on LLaMA / LLaMA‑2 / LLaMA‑3 models under 2‑ and 3‑bit settings show that FlexibleLLM significantly outperforms both finetuning-free and many finetuning-based baselines in perplexity and zero-shot accuracy, while using substantially lower computation cost. The results validate that more flexible bit allocation, better outlier handling, and cross-layer error correction lead to more robust and efficient low-bit quantization.

**Strengths:**

- The paper identifies key limitations in existing PTQ methods, including discrete outliers, attribute outliers, and cross-layer error accumulation, making the motivation very clear.

- Introducing SBGBS allows adaptive, fractional bit allocation at the block level, improving precision without full finetuning. DOSA and LFD further enhance robustness by handling discrete outliers and mitigating cross-layer errors.

- Evaluations span LLaMA, LLaMA‑2, and LLaMA‑3 models, under extremely low-bit settings (2-bit and 3-bit), demonstrating consistent improvements in perplexity and zero-shot tasks.

- FlexibleLLM is weight-only and finetuning-free, achieving better accuracy than many finetuning-based methods while keeping computational cost low, making it practical for large-scale LLM deployment.

**Weaknesses:**

- Combining SBGBS, DOSA, and LFD involves multiple stages and careful bookkeeping, increasing implementation difficulty.

- Block-level bit search and Hadamard-based outlier suppression may require tuning for different models or layers.

- Although finetuning-free, some steps like block-level greedy bit search and Hadamard transforms introduce extra computation compared to simpler PTQ methods.

- While the paper emphasizes the advantage of being finetuning-free, in practice the cost of light finetuning is often acceptable even for large models.

- The baselines used for comparison are relatively outdated and do not include more recent and stronger methods such as OSTQuant[1] or SpinQuant[2], which limits the significance of the claimed improvements.

- Anonymous code links seem to be inaccessible.

[1] Hu, Xing, et al. "Ostquant: Refining large language model quantization with orthogonal and scaling transformations for better distribution fitting." arXiv preprint arXiv:2501.13987 (2025).

[2] Liu, Zechun, et al. "Spinquant: Llm quantization with learned rotations." arXiv preprint arXiv:2405.16406 (2024).

**Questions:**

Please refer to the weaknesses above.

---

> ### Author Response · Authors · 2025-12-04
> **Author Responses (1/n)**
>
> > **W1**: Combining SBGBS, DOSA, and LFD involves multiple stages and careful bookkeeping, increasing implementation difficulty.
>
> We highly appreciate the concerns raised by the reviewers. However, we would like to add that although FlexibleLLM involves multiple stages, it still maintains strong robustness and implementability. Specifically, we have conducted hyperparameter experiments under the 2-bit quantization scenario for LLaMA-7B during the rebuttal stage:
>
> **Table 1: Influence of hyperparameter settings for SBGBS**
>
> | alapha | WikiText2 perplexity | Zero-shot average accuracy |
> |:------:|:--------------------:|:--------------------------:|
> |   1    |        9.02          |           50.64           |
> |   2    |        8.82          |           51.02           |
> |   3    |        8.85          |           51.19           |
> |   4    |        9.21          |           50.26           |
>
>
>
> **Table 2: Influence of hyperparameter settings for DOSA**
>
> | lamada1 | lamada2 | WikiText2 perplexity | Zero-shot average accuracy |
> |:-------:|:-------:|:--------------------:|:--------------------------:|
> |    1    |   1.0   |        9.76          |           48.94           |
> |    1    |   1.1   |        9.01          |           50.16           |
> |    1    |   1.2   |        8.82          |           51.02           |
> |    1    |   1.5   |        8.62          |           50.58           |
> |    1    |   2.0   |        8.73          |           50.87           |
>
>
> |   L   |   R   | WikiText2 perplexity | Zero-shot average accuracy |
> |:-----:|:-----:|:--------------------:|:--------------------------:|
> |  0.8  |  0.2  |        8.82          |           51.02           |
> |  0.7  |  0.3  |        8.99          |           50.54           |
> |  0.6  |  0.4  |        8.94          |           50.47           |
> |  0.5  |  0.5  |        9.24          |           49.66           |
>
>
> The experimental results demonstrate that different parameter settings of different module have little impact on the experimental outcomes, so users do not need to perform complex manual adjustments to its hyperparameters.
>
>
>
> > **W2**: Block-level bit search and Hadamard-based outlier suppression may require tuning for different models or layers.
>
>
>
> We sincerely thank the reviewers for their attention to this issue. We would like to clarify that block-wise bit search and Hadamard-based outlier suppression do not require gradient descent-dependent fine-tuning for different models or layers. **Specifically**, Block-wise bit search is a greedy principle-based iterative computation algorithm, and the Hadamard matrix in the Hadamard-based outlier module is constructed recursively. Both modules are independent of gradient descent for additional fine-tuning.
>
>
>
> > **W3**: Extra computation introduced by block-level greedy bit search and Hadamard transforms operations.
>
>
> We are grateful for the reviewer’s detailed comments. We would like to clarify that the additional computations introduced by these two modules are practically negligible when compared to fine-tuning based approaches. Meanwhile, we have also conducted comparative experiments to evaluate the **time overhead** of the block-wise bit search and the Hadamard-based outlier suppression module throughout the entire quantization process:
>
>
>
> | Component                    | Llama2-7B | Llama2-13B | Llama2-70B |
> |-----------------------------|-----------|------------|------------|
> | Hadamard matrix construction | 1.2min    | 2.9min     | 15.1min     |
> | Block-level bit search       | 9.2min   | 13.3min    | 64.2min    |
>
>
>
> More precisely, hadamard matrix construction and block-level bit search only need **1.2 minutes** and **9.2 minutes** in Llama1-7B. However, the fine-tuning phase of AffineQuant increases the total runtime of the quantization process by about **7 hours** under the same quantization scenario (more details in Sec 4.4 of the submission paper). Overall, FlexibleLLM delivers more excellent quantization performance with relatively low additional resource overhead.
>
>
> > **W4**: While the paper emphasizes the advantage of being finetuning-free, in practice the cost of light finetuning is often acceptable even for large models.
>
> We sincerely thank the reviewers for pointing out this point. We would like to clarify that for LLMs, the fine-tuning costs of most existing fine-tuning based methods are still non-negligible. We have further reported the computational cost of the fine-tuning based methods AffineQuant and OminiQuant on 80GB A800GPUs:
>
> | Method      | Llama2-7B | Llama2-13B | Llama2-70B |
> |-------------|-----------|------------|------------|
> | AffineQuant | 7.11h     | 11.17h     | 23.92h     |
> | OminiQuant  | 7.27h     | 10.68h     | 24.88h     |
>
> The results show that the fine-tuning overhead of commercially available OmniQuant and AffineQuant is much more huger compared with the process of Hadamard matrix construction and block-level bit search above.

---

> ### Author Response · Authors · 2025-12-04
> **Author Responses (2/n)**
>
> > **W5**: The selected baseline methods are relative outdated.
>
> We highly appreciate the reviewer’s insightful comments. We have compared our method with the new state-of-the-art approaches such as spinquant during the rebuttal stage:
>
> | Method       | Llama1-7B | Llama1-7B | Llama2-7B | Llama2-7B |
> |--------------|-----------|-----------|-----------|-----------|
> |              | **WikiText2 perplexity** | **Zero-shot average accuracy** | **WikiText2 perplexity** | **Zero-shot average accuracy** |
> | OSTQuant     | 15.31     | 46.32     | 45.54     | 44.64     |
> | Spinquant    | 25.39     | 44.54     | 53.17     | 45.20     |
> | FlexibleLLM  | **8.82**      | **51.02**     | **8.70**      | **52.70**     |
>
>
>
> The above experimental results demonstrate that FlexibleLLM still maintains a significant advantage over these methods in low-bit quantization scenarios. It is worth noting that the poor performance of these recent methods in low-bit quantization regimes can be attributed to their activation-based rotation operations, which disrupt the intrinsic relative distribution of the weight matrix, whereas our NOR module tackles this challenge well from a denoising perspective.
>
>
> > **W6**: Anonymous code links seem to be inaccessible.
>
> We thank the reviewers for pointing out this point. The initial inaccessibility was likely due to a temporary network issue on the code platform. We have rechecked the anonymous code links, and they are now fully accessible.

---

### Official Review · Reviewer_LcMU · 2025-11-01

**Soundness:** 4
**Presentation:** 3
**Contribution:** 3
**Rating:** 6
**Confidence:** 4

**Summary:**

The paper proposes FlexibleLLM, a finetuning-free, weight-only Post-Training Quantization (PTQ) framework designed to make low-bit quantization of Large Language Models (LLMs) more accurate, flexible, and efficient, especially for deployment on resource-limited devices. It rethinks the nature of outliers in quantization, builds a modular finetuning-free framework with adaptive and fractional bit-width control, and sets a new benchmark for low-bit, high-accuracy LLM quantization.

**Strengths:**

1. Contribution of the Paper：The paper makes an original and technically solid contribution to the low - bit quantization of LLMs. It introduces FlexibleLLM, a finetuning - free PTQ framework that holistically addresses key limitations of existing methods through three modules.

2. Modular design and High integration：FlexibleLLM use Self-Adaptive Block-Level Bit Search (SBGBS) module, Discrete Outlier Suppression and Awareness (DOSA) module, and Layer-Level Feedback Denoising (LFD) module to factilitate integtation into LLMs.

3. Theory and Concept:The work is notable for its conceptual originality. It offers a new theoretical perspective on outliers by distinguishing between discrete, clustered, and “attribute” outliers.It also models the cumulative effects of quantization noise across layers.

4. Methodology: The methodology is rigorous and empirically well - supported across multiple model families. It delivers strong performance improvements over SOTA baselines with minimal computational overhead.

**Weaknesses:**

1. Lack of Direct Validation:The experimental results do not directly validate these specific theoretical claims.

2. Impressive but Unclear Performance Improvements: The reported performance improvements are impressive but could be due to general optimization and calibration effects rather than the specific theoretical mechanisms proposed.

3. Missing Controlled Experiments:There are no controlled experiments that:

-  Isolate the influence of “attribute outliers” on quantization error.

- Visualize or quantify the dual distribution of outliers.

- Explicitly measure cross-layer noise accumulation before and after the LFD module.

**Questions:**

1. The paper claims that outliers exhibit both discrete and clustered distributions. Could the authors present quantitative or visual evidence (e.g., density plots or clustering metrics) supporting this claim?

2. Can the authors show how quantization noise propagates before and after applying LFD? For instance, is there a measurable reduction in activation variance or output deviation at deeper layers?

---

> ### Author Response · Authors · 2025-12-04
> **Author Responses (1/n)**
>
> > **W1**: Lack of Direct Validation:The experimental results do not directly validate these specific theoretical claims.
>
> We sincerely thank the reviewers for pointing out this point. We wish to illustrate that we have provided intuitive visualization results in the submitted paper to validate the theoretical claims of this work.
> - **Specifically**, for the outlier distribution, we have visualized the outlier distribution in the attention output layer of LLaMA2-7B’s 16th layer in **Figure 1(b) of the paper**, and the revealed distribution patterns are consistent with our proposed theory.
> - **Additionally**, in **Figure 1(c) of the paper**, we have evaluated and demonstrated the differences in outlier attribute magnitudes across layers and the accumulation of quantization noise. The highlighted distribution regularities align with our theoretical proofs, further indicating that outliers are the dominant factor determining quantization noise.
>
>
> > **W2**: Impressive but Unclear Performance Improvements: The reported performance improvements are impressive but could be due to general optimization and calibration effects rather than the specific theoretical mechanisms proposed.
>
>
> We are grateful for the reviewer’s detailed comments. We wish to clarify that all modules in our method are designed and implemented based on the proposed theoretical mechanisms. **Table 1** summarizes the correspondence between each module and its associated theoretical mechanism, and further reports the perplexity increase observed when each module is individually removed in the 2-bit Llama1-7B quantization setting. In addition, **Section 4.3 of the paper** presents detailed ablation studies for each module, and the results further validate the effectiveness of the proposed theoretical mechanisms.
>
> **Table 1: Ablation study and mapping between theoretical mechanisms and modules**
> | Module | Corresponding Theoretical Mechanism                                        | Δ WikiText2 perplexity  |
> |--------|----------------------------------------------------------------------------|-------------------------|
> | SBGBS  | Clustered-distribution outliers in planar space                            | 0.55 ↑                  |
> | NOR    | Discrete numerical-magnitude outliers in planar space                      | 3.10 ↑                  |
> | HGC    | Discrete intrinsic-attribute outliers in planar space                       | 2.00 ↑                  |
> | LFD    | Layer-wise evolution of outliers in multi-dimensional stereoscopic space    | 1.17 ↑                  |
>
>
>
>
>
>
> > **W3,W1**: Missing direct validation of specific theoretical claims and controlled experiments.
>
>
> We are grateful for the reviewers’ constructive feedback. In response to your questions, we have performed additional comparative experiments.
>
> - **First**, we evaluated the impact of not addressing attribute outliers on the quantization error for LLaMA2-7B under the 2-bit quantization setting. The results are shown in form of Table below:
>
> | Layer                |   1    |   8    |   16   |   24   |   31    |
> |:--------------------:|:------:|:------:|:------:|:------:|:-------:|
> | W Attribute_outlier  | 415.6  | 421.6  | 464.0  | 564.2  | 1472.8  |
> | W/O Attribute_outlier| 356.4  | 361.7  | 386.3  | 454.2  | 947.6   |
>
> This outcome further demonstrates that adequate attention to attribute outliers can mitigate quantization error to a certain extent and reduce the amplitude of exponential jump in the final-layer quantization error to some degree.
>
> - **Next**, concerning the dual distribution of outliers (discrete and clustered) you raised, a visualization has been included in **Figure 1(b) of the paper**.
>
> - **Finally**, we explored the accumulation of quantization noise with and without the LFD module, and a corresponding visualization is provided in **Figure 1(c) of the paper**.
>
> - **Overall**, these comparative experiments have further validated the reliability of our proposed theoretical mechanism. We also hope this theoretical mechanism facilitating future researchers in constructing more efficient quantization algorithms.

---

### Official Review · Reviewer_ka1L · 2025-11-01

**Soundness:** 3
**Presentation:** 3
**Contribution:** 2
**Rating:** 6
**Confidence:** 2

**Summary:**

This paper introduces FlexibleLLM, a finetuning-free, weight-only PTQ framework for large language models. It combines three key components: SBGBS for adaptive bitwidth allocation， DOSA for outlier suppression using a Hadamard transform and Hessian-aware calibration, and LFD for cross-layer error correction. The method targets both clustered and discrete outliers and reports consistent improvements.

**Strengths:**

- The paper is well-motivated: it focuses on real problems—mixed outlier types and error accumulation—that most PTQ works overlook.
- The integration of SBGBS, DOSA, and LFD forms a complete and coherent pipeline, not just a single heuristic.
- Experimental coverage across LLaMA and Qwen models shows strong potential.

**Weaknesses:**

- The inverse-Hessian correction might be heavy for large models. How is $H^{-1}$ approximated?
- Many recent PTQ works (e.g., SpinQuant, QuaRot) also perform rotation or error compensation. The novelty boundary between DOSA and these methods should be better articulated.

**Questions:**

- Weakness 1&2
- What’s the absolute per-token latency and throughput with and without DOSA and LFD on A100/H100?
-  Can you show empirical per-layer error amplification curves after applying LFD?

---

> ### Author Response · Authors · 2025-12-04
> **Author Responses (1/n)**
>
> > **W1,Q1(1)**: The inverse-Hessian correction might be heavy for large models. How is it approximated?
>
> We appreciate the reviewer’s detailed comments. We would like to clarify that FlexibleLLM indeed adopts an approximation of the inverse Hessian to avoid the substantial computational cost associated with computing it directly. Specifically, we follow the result from OBS [1], which approximates the inverse Hessian using the expression ${H^{ - 1}} = {\left( {X{X^T}} \right)^{ - 1}}$. This approximation eliminates the heavy computational burden of explicitly computing the Hessian matrix, and its stability and validity have been rigorously established in OBS [1].
>
> [1] Hassibi, Babak, David G. Stork, and Gregory J. Wolff. "Optimal brain surgeon and general network pruning." IEEE international conference on neural networks. IEEE, 1993.
>
>
> > **W2,Q1(2)**: The difference between DOSA with some related ratation based PTQ methods.
>
> We appreciate the reviewer’s constructive comments regarding these recent methods. We would like to clarify that although these approaches share certain similarities with DOSA, there are also fundamental differences. These differences can be summarized as follows:
>
> - **DOSA adopts a more diverse perspective to detect and handle outliers**: The primary distinction is that recent methods typically define and address outliers solely from the perspective of numerical magnitude. In contrast, DOSA considers both numerical magnitude and intrinsic properties when identifying and processing outliers. This allows DOSA to effectively capture elements whose numerical magnitude appear normal but nonetheless exert a substantial influence on quantization performance, which we refer to in the paper as “attribute outliers”.
>
> - **DOSA requires very little computational cost**: For numerical outliers, methods like SpinQuant rely on fine-tuning, whereas FlexibleLLM does not. SpinQuant requires substantial GPU resources, and quantizing the LLaMA2-70B model takes more than one day on multiple A100 GPUs. DOSA avoids this heavy computational cost.
>
> - **DOSA explicitly accounts for noisy rotation matrices**: Specifically, DOSA actively addresses the noise contained in randomly generated Hadamard matrices. In comparison, although existing rotation-based methods (Quarot, SpinQuant, FlatQuant) also use Hadamard matrices, their usage largely remains at the stage of random generation, without further processing to mitigate the noise and potential adverse effects introduced by this randomness.
>
> > **Q2**: absolute per-token latency and throughput with and without DOSA and LFD on A100.
>
> We are grateful for the detailed comments and insightful suggestions provided by the reviewers. In response to your inquiries, we conducted additional evaluations on the inference speed variations with the DOSA and LFD modules enabled or disabled, using an A100 GPU. Below are the detailed results:
>
> | Model        | W DOSA (W LFD) | W/O DOSA     | W/O LFD       |
> |--------------|----------------|--------------|---------------|
> | **Llama 1-7B**  | 81.99 token/s  | 85.78 token/s | 81.99 token/s |
>
>
> The experimental results demonstrate that enabling or disabling the LFD has no impact on inference speed. This is because these modules are optimization algorithms implemented during the quantization process and they do not affect the inference speed of the quantized model. Meanwhile, we can observe a slight decrease in the inference performance of the quantized model after enabling the DOSA module, which can be attributed to the use of a Hadamard transform matrix in the DOSA module. However, since all elements of the Hadamard matrix are ±1, its multiplication operations only involve addition and subtraction. This characteristic ensures that the overhead introduced by the NOR module is almost negligible during model inference.
>
>
>
>
> > **Q3**: Can you show empirical per-layer error amplification curves after applying LFD?
>
> We are grateful to the reviewers for identifying this key point. The empirical curves illustrating the layer-wise error amplification with and without the application of LFD have been presented in **Figure 1(c) of our paper** of our submission paper. Specifically, "W/O LFD" refers to the case without the LFD module, and "W/ LFD" corresponds to the case with the LFD module applied.

---

### Official Review · Reviewer_yusG · 2025-11-01

**Soundness:** 3
**Presentation:** 3
**Contribution:** 3
**Rating:** 4
**Confidence:** 4

**Summary:**

The paper proposes FlexibleLLM, a finetuning-free, weight-only PTQ framework for LLMs. It tackles (i) clustered vs. discrete outliers and (ii) cross-layer error accumulation via three components:

- SBGBS: a self-adaptive, block-level greedy bit search that allocates fractional bit widths (e.g., 2.1 bits) based on importance.

- DOSA: a two-part module—(NOR) numerical outlier reduction via Hadamard transforms, and (HGC) a Hessian-guided mechanism to address “attribute outliers.”

- LFD: layer-level feedback/denoising to mitigate activation-noise propagation with an error-aware objective.

Experiments show improvements over several PTQ baselines on some metrics and models; the method is presented clearly with ablations.

**Strengths:**

1. The framework is easy to follow; writing is organized and readable.

2. Reasonable benchmarks and ablations; measurable gains on some tasks.

3. Attention to both numerical and “attribute” outliers: Explicit handling is practically relevant.

**Weaknesses:**

1. Limited novelty / incremental combinations: Each module largely extends known ideas with modest variations:

- SBGBS: Importance-guided mixed-precision allocation is well-studied; the greedy search and block granularity feel incremental.

- NOR (in DOSA): Hadamard transforms for outlier suppression/energy spreading are mature and previously explored.

- HGC (in DOSA): Using Hessian information (or approximations) to guide quantization or error reconstruction is established.

- LFD: Using output-error/activation-noise–aware objectives echoes prior PTQ works (e.g., QDrop-style criteria and related error-aware calibration).

Overall, the contribution reads more as a package of small updates than a distinct conceptual advance.

2. Many moving parts: Four submodules and several hyperparameters risk over-engineering, making it harder to attribute gains to any single idea and increasing deployment complexity.

3. Ablation depth: While ablations exist, they don’t isolate what is truly new versus what prior techniques would already deliver under matched calibration/tuning.

4. Theory claims: The “new theoretical analysis of outliers” is not clearly distinguished from existing analyses; formal novelty remains unclear.

**Questions:**

Do gains persist across different seeds, calibration set choices/sizes, and task distributions (generation vs. perplexity vs. alignment tasks)? Any evidence of overfitting to the calibration set despite being “finetuning-free”?

---

> ### Author Response · Authors · 2025-12-04
> **Author Responses (1/n)**
>
> > **W1**: Limited novelty / incremental combinations.
>
> We sincerely appreciate the reviewer’s thoughtful comments. We wish to clarify that the methodology innovation is only one aspect of the broader set of contributions in this paper. To sum up, our work achieves innovations in three aspects: **motivation**, **theory**, and **methodology**:
>
> - **Motivation innovation**: Compared with recent latest quantization approaches(Quarot, SpinQuant, FlatQuant, OSTQuant, GPTAQ, SLIM-LLM, Q-Drop), **our motivation innovation lies in exploring a flexible fractional-bit quantization framework that aims to better balance device storage constraints and model performance**. In practice, many consumer-level users face strict storage limits, which often become deployment bottlenecks. For example, a 2-bit quantized model may fit into memory, while a 3-bit model can already exceed the budget. Therefore, we aim to offer flexible fractional-bit quantization so that users can choose configurations that best match their device-specific storage constraints.
> - **Theory innovation**: In terms of theoretical contribution, our work differs fundamentally from recent quantization studies. Specifically, **Our work is among the few that directly introduce a multi-scale analytical framework which systematically characterizes outliers and the quantization errors they induce from both local and global perspectives**. This framework jointly considers the magnitude and intrinsic attributes of outliers and offers a comprehensive view of their distribution and evolution, thereby revealing the key bottlenecks that limit quantization performance. **In contrast**, most recent works include only limited formula-based analyses aimed at explaining specific implementation details, rather than probing the fundamental bottlenecks of quantization.
> - **methodology innovation**: The methodology innovation in our work are directly inspired by the motivation and theory innovations described above. **Specifically** in the local planar space, the SBGBS and DOSA modules comprehensively handle binary-distribution outliers from both the perspectives of numerical magnitude and intrinsic attributes. In the global planar space, the LFD module adaptively perceives the importance of each layer and performs dynamic quantization error compensation based on layer-wise calibration factors and full-precision activations. The synergy among these components enables more effective and flexible low-bit quantization. **In the following**, we describe in detail how the components of FlexibleLLM differ from recent similar quantization techniques.
>   - **SBGBS**: Most existing importance-guided mixed-precision methods have two main limitations: they either rely on static sensitivity metrics inspired by pruning methods such as SparseGPT, or they depend solely on the magnitude of individual weights, both of which ignore the interaction between sensitivity and actual quantization error. **In contrast**, SBGBS jointly considers static weight sensitivity and dynamic quantization error, and further adopts a greedy optimization strategy that improves interpretability and promotes convergence toward a near-global optimum.
>
>   - **NOR**: Recent Hadamard-based rotation methods such as Quarot still rely on randomly generated Hadamard matrices and do not explicitly address the noise they introduce into the relative distribution of weight elements. **In contrast**, NOR explicitly optimizes these matrices via feature engineering, local K-means clustering, and a two-level sorting strategy to achieve a denoising effect (**refer to Appendix E of the paper**), thereby mitigating the negative impact of noisy Hadamard rotations on the weight distribution.
>
>   - **HGC**: Most existing Hessian-based methods such as GPTQ only indirectly utilize Hessian information as a computational proxy to simplify some technical implementation details. **In contrast**, HGC leverages Hessian information directly as a core indicator for outlier assessment. Guided by hessian information directly, it further explores and addresses the "attribute outliers" defined in this work, which refer to elements that exhibit normal numerical values but have considerable influences on the quantization of practical impacts.
>
>   - **LFD**: Existing output error-aware methods typically rely on gradient descent-based fine-tuning for calibration, which requires substantial computational overhead during the quantization process. **In contrast**, the LFD module is designed to be resource-friendly. It markedly reduces the computational resource demands of quantization through a mathematically derived one-shot closed-form calibration scheme combined with hardware-aware co-optimization strategy. As further demonstrated in **Sec. 4.4 of the paper**, our approach achieves significant savings in computational resources compared with fine-tuning based calibration methods.

---

> > ### Author Response · Authors · 2025-12-04
> > **Author Responses (3/n)**
> >
> > > **W4**: The “new theoretical analysis of outliers” is not clearly distinguished from existing analyses; formal novelty remains unclear.
> >
> > We appreciate the reviewer’s valuable comments. We would like to clarify that our theoretical analysis begins directly from the core challenge that limits quantization performance, namely outliers, and then proposes a comprehensive multi-scale framework for analyzing outliers and the quantization errors they introduce. Specifically:
> > - **First**, we reveal the distribution patterns and evolution trends of outliers from both local and global perspectives.
> > - **Then**, our analysis provides a layer-level characterization of quantization error behaviors that are closely related to outliers, through which we identify and demonstrate the exponential growth of quantization error in the final layer of large language models.
> > - **Finally**, it is worth noting that few existing works examine quantization from the viewpoint of the fundamental performance bottleneck; most theoretical derivations of recent works focus primarily on justifying certain implementation details rather than analyzing the underlying limiting factors.

---

> > ### Author Response · Authors · 2025-12-04
> > **Author Responses (4/n)**
> >
> > > **Q1(1)**: Do gains persist across different seeds, calibration set choices/sizes, and task distributions ?
> >
> > We appreciate the reviewer’s careful comments and valuable suggestions. To address your concerns, we further explored and discussed the effects of seed selection, calibration dataset size, and task variation on the performance of FlexibleLLM:
> >
> > - **First**, we randomly selected five seeds on both the LLaMA1-7B and LLaMA2-7B models and reported the corresponding standard deviations:
> >
> >
> > |                | Llama1-7B | Llama1-7B | Llama2-7B | Llama2-7B |
> > |----------------|-----------|-----------|-----------|-----------|
> > |                | mean      | std       | mean      | std       |
> > | **WikiText2 perplexity**      | 8.82      | 0.28      | 8.70      | 0.14      |
> >
> >
> >  The above results show that our method consistently maintains its performance advantages across different seeds, indicating strong robustness. We would also like to emphasize that all results reported in the original paper are the averages over five independent runs with different random seeds.
> >
> >
> >
> >
> >
> >
> > - **Next**, we conducted experiments on both the LLaMA1-7B and LLaMA2-7B models using calibration datasets of sizes 32, 64, 128, 256, and 512. As shown in **Table 6**, even when using only 32 calibration samples, our method still outperforms most existing approaches that rely on 128 calibration samples **(more details refer to Sec. 4.2 of the submission paper)**. The results also indicate that larger calibration datasets generally lead to improved quantization performance. However, considering the trade-off between computational cost and performance, a dataset size of 128 appears to be the most suitable choice. Overall, FlexibleLLM maintains its performance advantages and demonstrates strong robustness across different calibration dataset sizes.
> >
> > **Table 6: Quantization Performance  under different calibration dataset sizes**
> >
> > | Calibration size |  Llama1-7B  |  Llama2-7B  |
> > |:-----------:|:------:|:------:|
> > |     32      |  9.20  |  9.31  |
> > |     64      |  8.99  |  9.08  |
> > |    128      |  8.82  |  8.70  |
> > |    256      |  8.57  |  8.67  |
> > |    512      |  8.61  |  8.73  |
> >
> >
> > - **Finally**, during the rebuttal stage, we supplemented additional evaluations of the models quantized by FlexibleLLM on new reasoning and mathematical tasks:
> >
> > |  Bit  |   Model    |   method    | lambada_openai | lambada_standard | mathqa | MMLU_avg | WikiText2 perplexity |
> > |:-----:|:----------:|:-----------:|:--------------:|:----------------:|:------:|:--------:|:--------------------:|
> > | 2 bit | Llama2-7B  |   GPTQ      |      0.00      |       0.00       | 21.12  |  21.89   |        60.45         |
> > | 2 bit | Llama2-7B  | OmniQuant   |     4.02      |      1.78       | 23.74  |  23.71   |        11.06         |
> > | 2 bit | Llama2-7B  | AffineQuant |     6.57      |      3.43       | 22.68  |  23.67   |        10.87         |
> > | 2 bit | Llama2-7B  |    **Ours**     |     **46.56**      |      **38.73**       | **24.08**  |  **23.93**   |         **8.70**         |
> > | 2 bit | Llama1-7B  |   GPTQ      |      0.00      |       0.00       | 21.97  |  21.56   |       152.31         |
> > | 2 bit | Llama1-7B  | OmniQuant   |     23.89      |      15.78       | 23.32  |  23.45   |         9.71         |
> > | 2 bit | Llama1-7B  | AffineQuant |     22.35      |      15.68       | 22.17  |  **23.51**   |        13.51         |
> > | 2 bit | Llama1-7B  |    **Ours**     |     **38.02**      |      **29.14**       | **23.94**  |  23.48   |         **8.82**         |
> >
> > The experimental results show that our method consistently outperforms existing approaches on these tasks. In the original paper, we also reported in **Sec. 4.2 of the paper** the performance of the FlexibleLLM-quantized models on a variety of zero-shot benchmarks, including Winogrande, PIQA, ARC-c, ARC-e, and HellaSwag. These results similarly indicate that FlexibleLLM maintains its performance advantages across different task distributions.
> >
> >
> > > **Q1(2)**: Any evidence of overfitting to the calibration set despite being “finetuning-free”?
> >
> > We appreciate the reviewer’s attention to this question. We would like to clarify that overfitting is generally defined as a situation where a model performs well on the calibration (or training) dataset but significantly worse on the test dataset, a phenomenon that becomes more likely when the calibration set is very small. However, as shown in **Table 6 above**, even when the calibration dataset size is reduced to 32 samples, the performance of the quantized models does not collapse. This indicates that FlexibleLLM does not exhibit overfitting with respect to the calibration dataset and maintains good robustness.

---

> ### Author Response · Authors · 2025-12-04
> **Author Responses (2/n)**
>
> > **W2(1)(3)**: Four submodules and several hyperparameters risk over-engineering, increases deployment complexity.
>
> We fully understand and appreciate the reviewer’s concern. To address this issue, we conducted additional hyperparameter ablation studies under the 2-bit quantization scenario for LLaMA1-7B during the rebuttal stage:
>
>  **Table 1: Influence of hyperparameter settings for SBGBS**
>
> | alapha | WikiText2 perplexity | Zero-shot average accuracy |
> |:------:|:--------------------:|:--------------------------:|
> |   1    |        9.02          |           50.64           |
> |   2    |        8.82          |           51.02           |
> |   3    |        8.85          |           51.19           |
> |   4    |        9.21          |           50.26           |
>
>
>
> **Table 2: Influence of hyperparameter settings for DOSA**
>
> | lamada1 | lamada2 | WikiText2 perplexity | Zero-shot average accuracy |
> |:-------:|:-------:|:--------------------:|:--------------------------:|
> |    1    |   1.0   |        9.76          |           48.94           |
> |    1    |   1.1   |        9.01          |           50.16           |
> |    1    |   1.2   |        8.82          |           51.02           |
> |    1    |   1.5   |        8.62          |           50.58           |
> |    1    |   2.0   |        8.73          |           50.87           |
>
>
> |   L   |   R   | WikiText2 perplexity | Zero-shot average accuracy |
> |:-----:|:-----:|:--------------------:|:--------------------------:|
> |  0.8  |  0.2  |        8.82          |           51.02           |
> |  0.7  |  0.3  |        8.99          |           50.54           |
> |  0.6  |  0.4  |        8.94          |           50.47           |
> |  0.5  |  0.5  |        9.24          |           49.66           |
>
>  The above results indicate that variations in the hyperparameter settings of each module have only minor impact on performance, demonstrating strong robustness.
>
>
>
>
> > **W2(2)W3**: Ablation depth: While ablations exist, they don’t isolate what is truly new versus what prior techniques would already deliver under matched calibration/tuning.
>
> We appreciate the reviewer’s attention to this question. To further address your concern, we conducted additional and more detailed ablation studies during the rebuttal stage, directly comparing NOR and LFD module of FlexibleLLM with its related techniques in terms of performance and resource consumption:
>
>
>   **Table 3: Comparison of Quarot and NOR on Llama1-7B and Llama1-13B**
> |  Bit  |   Model    | Method | lambada_openai | lambada_standard | mathqa | MMLU_avg | WikiText2 perplexity |
> |:-----:|:----------:|:------:|:--------------:|:----------------:|:------:|:--------:|:--------------------:|
> | 2 bit | Llama1-7B  | Quarot |     12.74      |      11.68       | 20.46  |  **23.76**   |        18.07         |
> | 2 bit | Llama1-7B  |  NOR   |     **19.86**      |      **17.26**       | **22.65**  |  23.18   |        **15.19**         |
> | 2 bit | Llama1-13B  | Quarot |      13.56      |       10.11       | 21.40  |  23.94   |        16.15         |
> | 2 bit | Llama1-13B  |  NOR   |     **21.02**      |      **11.57**       | **23.26**  |  **24.40**   |        **12.53**         |
>
>  **Table 4: Comparison of GPTAQ and LFD on Llama1-7B and Llama1-13B**
> |  Bit  |   Model    | Method | lambada_openai | lambada_standard | mathqa | MMLU_avg | WikiText2 perplexity |
> |:-----:|:----------:|:------:|:--------------:|:----------------:|:------:|:--------:|:--------------------:|
> | 2 bit | Llama1-7B  | GPTAQ |      12.15      |       11.42       | 21.26  |  22.76   |        12.91         |
> | 2 bit | Llama1-7B  |  LFD   |     **14.92**      |      **13.06**       | **22.02**  |  **23.05**   |        **12.12**         |
> | 2 bit | Llama1-13B  | GPTAQ |     16.62      |      16.84       | 21.65  |  23.08   |        10.81         |
> | 2 bit | Llama1-13B  |  LFD   |     **17.79**      |      **18.37**       | **22.38**  |  **23.75**   |        **10.33**         |
>
> **Table 5: Comparison of GPU memory and quantization time per layer of Llama3-8B and Llama3-70B**
> | Method     | Llama3-8B Time | Llama3-8B Memory      | Llama3-70B Time | Llama3-70B Memory        |
> |------------|----------------|------------------------|-----------------|---------------------------|
> | GPTAQ      | **45.9s**          | 25.86GB                | **140.2s**          | 63.13GB                   |
> | GPTAQ(ALL_TO_CPU) | 55.58s         | 12.38GB                | 170.4s          | 35.28GB                   |
> | LFD        | 46.5s          | **10.54GB**                | 141.4s          | **30.85GB**                   |
>
>
>
>
> The results indicate that the modules of FlexibleLLM generally achieve notable performance improvements and fewer computational resource consumption compared with recent similar methods. This further highlights both the technical novelty and the resource efficiency of the components in FlexibleLLM.

---

### Meta-Review · Area_Chair_TPN8 · 2026-01-01

**Summary:**

The reviewers agree that the paper targets a highly relevant problem in post-training quantization for large language models. They acknowledge that the motivation is clear and that the proposed framework is practically oriented toward enabling low-bit deployment under strict resource constraints. However, several concerns significantly shape the overall evaluation.

The most critical issue involves the level of novelty and contribution. Some reviewers perceive the method as a set of modular integrations rather than a fundamentally new quantization strategy. One reviewer specifically argues that each major component strongly resembles prior work and therefore sees limited conceptual innovation. This raises doubts about whether the paper advances the state of the art in a meaningful way.

Another recurring concern relates to the connection between theory and empirical validation. The paper introduces theoretical claims regarding the nature of outliers and the propagation of quantization noise across layers, yet reviewers note that experimental evidence demonstrating these mechanisms remains indirect or insufficient. They would like to see controlled experiments that more clearly isolate and confirm the proposed theoretical insights.

Reviewers also highlight concerns about implementation complexity and practical deployment. Although the method is finetuning-free, it still requires several processing stages such as block-level bit searching and Hadamard-based transformations. Some reviewers believe that these steps may increase tuning difficulty and computational overhead, especially when scaled to larger models.

In addition, reviewers point out gaps in evaluation. They believe that the reliance on perplexity does not fully reflect modern LLM performance expectations and ask for broader reasoning benchmarks. They also note that comparisons against some of the most recent and strongest baselines are missing, which weakens the strength of the reported improvements.

Finally, reproducibility and transparency are a source of concern. One reviewer observed substantial similarity between the provided code and existing public implementations, and requested clearer clarification of what is newly introduced. Temporary inaccessibility of the code repository further reduced confidence in the reproducibility of the proposed approach.

Taken together, the reviewers find the topic timely and the goal important, but they express reservations regarding originality, empirical support for theoretical claims, engineering complexity, evaluation completeness, and code credibility. These concerns form the basis of the reviewers’ hesitation in recommending acceptance at this stage.

**Reviewer Concerns:**

The rebuttal has made a sincere effort to address multiple concerns, particularly those related to empirical validation and evaluation completeness. The authors expanded their experiments to include additional robustness results, more reasoning benchmarks, and direct comparisons with stronger recent baselines. They also provided more detailed explanations of module design choices and clarified code-related misunderstandings. These additions improve the clarity and technical soundness of the submission and demonstrate that the authors took the reviewer feedback seriously.

However, some of the core concerns remain insufficiently resolved. Most notably, the question of novelty still persists after the rebuttal. While the authors articulate theoretical motivations and provide arguments that their modules differ from existing approaches, the overall method continues to appear as a relatively complex combination of previously established techniques rather than a fundamentally new direction. The fact that multiple hyperparameters and procedural components are required, even if shown to be relatively insensitive, can be interpreted as further evidence that the contribution is layered on top of existing ideas rather than breaking new ground. Additionally, the explanations distinguishing the proposed modules from prior work sometimes become complicated and indirect, which paradoxically weakens the claim of conceptual originality.

Furthermore, some details that are positioned as methodological innovations, such as Hessian-based correction strategies or Hadamard transformations, are well-known elements in prior literature. Although the authors provide more justification, these aspects do not fully dispel the concern that the approach may lack a strong novel core. The code availability situation is understandable given institutional constraints, but because part of the originality debate hinges on implementation differences, a more transparent demonstration with unequivocally distinct experimental outcomes would be needed to fully eliminate doubts.

In summary, the rebuttal successfully addresses several secondary concerns, particularly in terms of evaluation breadth and presentation clarity. Nevertheless, the primary concern regarding the degree of novelty remains largely outstanding and continues to limit confidence in the strength of the contribution.

**Reviewer Scores:**

Given the nature of the concerns raised, especially those centered on fundamental questions of novelty and originality, I believe it is unlikely that the more critical reviewers would significantly change their scores even after further discussion. While the rebuttal provides additional experiments and clarifications that may positively influence perceptions of soundness and evaluation completeness, the core doubts regarding whether the proposed framework introduces a truly novel contribution remain largely unresolved. Because these concerns are foundational rather than gaps in clarity or missing results, they are less likely to be shifted through discussion alone.

Therefore, my expectation is that the reviewers who already viewed the contribution as incremental would probably maintain their current assessment.

---

### Decision · Program_Chairs · 2026-01-26

Reject